# Molecular basis for gating of cardiac ryanodine receptor explains the mechanisms for gain- and loss-of function mutations

Takuya Kobayashi [1,5], Akihisa Tsutsumi [2,5], Nagomi Kurebayashi [1], Kei Saito[3], Masami Kodama[1], Takashi Sakurai [1], Masahide Kikkawa [2], Takashi Murayama [1✉] & Haruo Ogawa [4✉]

Cardiac ryanodine receptor (RyR2) is a large $Ca^{2+}$ release channel in the sarcoplasmic reticulum and indispensable for excitation-contraction coupling in the heart. RyR2 is activated by $Ca^{2+}$ and RyR2 mutations are implicated in severe arrhythmogenic diseases. Yet, the structural basis underlying channel opening and how mutations affect the channel remains unknown. Here, we address the gating mechanism of RyR2 by combining high-resolution structures determined by cryo-electron microscopy with quantitative functional analysis of channels carrying various mutations in specific residues. We demonstrated two fundamental mechanisms for channel gating: interactions close to the channel pore stabilize the channel to prevent hyperactivity and a series of interactions in the surrounding regions is necessary for channel opening upon $Ca^{2+}$ binding. Mutations at the residues involved in the former and the latter mechanisms cause gain-of-function and loss-of-function, respectively. Our results reveal gating mechanisms of the RyR2 channel and alterations by pathogenic mutations at the atomic level.

[1] Department of Cellular and Molecular Pharmacology, Juntendo University Graduate School of Medicine, Tokyo, Japan. [2] Department of Cell Biology and Anatomy, Graduate School of Medicine, The University of Tokyo, Tokyo, Japan. [3] Department of Life Sciences, Graduate School of Arts and Sciences, The University of Tokyo, Tokyo, Japan. [4] Department of Structural Biology, Graduate School of Pharmaceutical Sciences, Kyoto University, Kyoto, Japan. [5]These authors contributed equally: Takuya Kobayashi, Akihisa Tsutsumi. ✉email: takashim@juntendo.ac.jp; haru@pharm.kyoto-u.ac.jp

Cardiac ryanodine receptor (RyR2) is a $Ca^{2+}$ release channel in the sarcoplasmic reticulum and plays a central role in cardiac muscle contraction[1,2]. In cardiac excitation–contraction coupling, $Ca^{2+}$ influx occurs through L-type voltage-dependent $Ca^{2+}$ channels in the transverse (T) tubule membrane activates RyR2 to release a large amount of $Ca^{2+}$, a process known as $Ca^{2+}$-induced $Ca^{2+}$ release (CICR)[3,4]. In human RyR2, nearly 300 pathogenic mutations have been reported as arrhythmogenic heart diseases, including catecholaminergic polymorphic ventricular tachycardia (CPVT)[5–8], and idiopathic ventricular fibrillation (IVF)[9–12]. Notably, these mutations divergently alter channel activity: mutations related to CPVT cause gain-of-function, whereas those for IVF lead to either gain- or loss-of-function[5–14].

RyR is a large tetrameric ion channel (~2.2 MDa), with each monomer composed of ~5000 amino acid residues and comprising a large N-terminal mushroom-like structure of the cytoplasmic domain and C-terminal transmembrane (TM) region[15–18]. The opening of RyR channels is initiated by binding of $Ca^{2+}$ to the $Ca^{2+}$-binding site, which was proposed at the interface between central and CTD domains by structural analysis using cryo-electron microscopy (EM)[19] and, later, validated by functional analysis using site-directed mutagenesis[20,21]. The mechanism of channel opening by $Ca^{2+}$ binding has been extensively studied by structural analysis of the channel at the open state[19,22–26]. These results demonstrated that $Ca^{2+}$-binding causes a large conformational change throughout the molecule via a large number of intra/inter-domain interactions, thereby the channel opens.

The activity of the RyR channels is modulated by various small molecules and associated proteins[27]. The RyR structures in complex with regulatory molecules, such as ATP, caffeine, FKBP12, or calmodulin, have also been determined, and the mechanisms by which these regulatory molecules modulate RyR have been extensively discussed[19,25,26]. In addition, the structures of three different gain-of-function mutants with mutations in the N-terminal domains have recently been reported[28,29]. In all structures, slight slippages in the inter-domain interactions around the mutated site occur, and both groups concluded that these slippages may lead to the increase in the open probability in the channel through inter-domain interactions.

Although many structures of RyR have been so far determined, the molecular mechanism of channel gating of RyR still remains largely obscured, due to a lack of our knowledge about the key interactions for the conformational change, and how and in what order these multiple interaction-networks interlock for the channel opening. In addition, all the open structures obtained to date contain additional ligands (ATP, caffeine, and/or PCB95), which help the opening of the channel but might make it difficult to discriminate conformational changes by $Ca^{2+}$. To precisely trace the $Ca^{2+}$-induced conformational changes, it is necessary to obtain an open structure with $Ca^{2+}$ alone.

In this study, we determine high-resolution closed and open structures of recombinant mouse RyR2. The open structure was successfully obtained by $Ca^{2+}$ alone. We perform a functional analysis of the RyR2 channels carrying mutations in amino acid residues involved in the observed interactions. Finally, to support our hypothesis of gating mechanism, we determine the structures of a loss-of-function mutant. We demonstrate two fundamental mechanisms for channel gating of RyR2. In the resting state without $Ca^{2+}$, the channel is stabilized in the closed state by multiple interactions close to the channel pore. Upon $Ca^{2+}$ binding, a series of interactions in the surrounding regions moves the S4-S5 linker outward to open the gate. Disruption of the former and the latter interactions causes gain-of-function and loss-of-function, respectively. Our results reveal mechanisms underlying channel opening upon $Ca^{2+}$ binding and explain how pathogenic mutations alter channel activity.

## Results

**Overall conformational changes in RyR2 associated with $Ca^{2+}$ binding.** Recombinant mouse RyR2, expressed and purified from HEK293 cells using FKBP12.6 affinity chromatography[30], formed homogeneous tetrameric channels (Supplementary Fig. 1a, b). To precisely trace the $Ca^{2+}$-induced RyR2 structural changes, we adopted high salt conditions, in which the channel activity was markedly enhanced by $Ca^{2+}$ alone but $Ca^{2+}$-dependent activation was preserved (Supplementary Fig. 1c–e)[31,32]. We performed the cryo-EM single-particle analysis in the presence of 1 mM EGTA and 100 μM $Ca^{2+}$ for closed and open states, respectively. Three-dimensional (3D) classification including focused classification analysis using the TM region revealed two and three classes in the presence of EGTA and $Ca^{2+}$, respectively (Supplementary Fig. 1f, g and Supplementary Table 1). No major differences were observed among the classes both in the closed and open states. Therefore, subsequent analysis was performed using the class with the highest resolution (before the final classification in the closed state and class 1 in the open state; Supplementary Table 1). The overall resolution of the final model was 3.3 Å and 3.45 Å for the closed and open states, respectively, and the local resolution for the TM region in the closed state was better than 2.9 Å (Supplementary Fig. 1f, g), which allowed us to build precise atomic models and identify specific residues important for channel gating (Supplementary Fig. 1h, i). In fact, it was confirmed that the pore was open only by the addition of 100 μM of $Ca^{2+}$ (Supplementary Fig. 2a), and moreover, the bound FKBP12.6 is clearly visible in both the closed and open states (Supplementary Fig. 1j, k). Conformational changes between closed and open states were large and spanned the entire molecule (Fig. 1a, b, Supplementary Fig. 2b–d and Supplementary movie 1). Changes associated with $Ca^{2+}$ binding were analyzed in three layers with different heights parallel to the membrane, i.e., the C-terminal domain (CTD), U-motif, and S4–S5 layers (Fig. 1c–e and Supplementary movie 1). Upon $Ca^{2+}$ binding, CTD rotated clockwise (viewed from the cytoplasmic side) (Fig. 1c), U-motif rotated clockwise toward the S2-S3 linker domain (Fig. 1d), and S6 leaned outward from the center to open the gate accompanied by rearrangements of S1–S4 helices and outward movement of the S4–S5 linker (Fig. 1e). The C-terminal side of the Central domain (C-Central), U-motif, CTD, and cytoplasmic S6 ($S6_{cyto}$) were tightly attached to each other in both closed and open states (Fig. 1f, Supplementary Fig. 2e, f and Supplementary movie 2) and rotated together 9.8° clockwise with respect to the axis approximately parallel to S6 (Fig. 1g). In contrast, the N-terminal side of the Central domain (N-Central) did not follow the rotation due to splitting the Central domain into two parts with G3987 as the pivot (Fig. 1f). Since the rotation axis was close to the $Ca^{2+}$-binding site, slight movements caused by $Ca^{2+}$ binding resulted in very large movements around S6 (Fig. 1g).

Given these series of movements, we hypothesized that the rotation of the C-Central/U-motif/$S6_{cyto}$/CTD complex upon $Ca^{2+}$ binding leads to two independent downstream pathways for channel pore opening; (1) the $S6_{cyto}$ movement leads to the outward leaning of S6; (2) the U-motif movement leads to sequential movements of the S2–S3 linker domain and TM segments, causing outward movement of the S4–S5 linker and creating a space where S6 can lean into (Fig. 1h). To prove this hypothesis, we performed detailed structural analysis and functional studies.

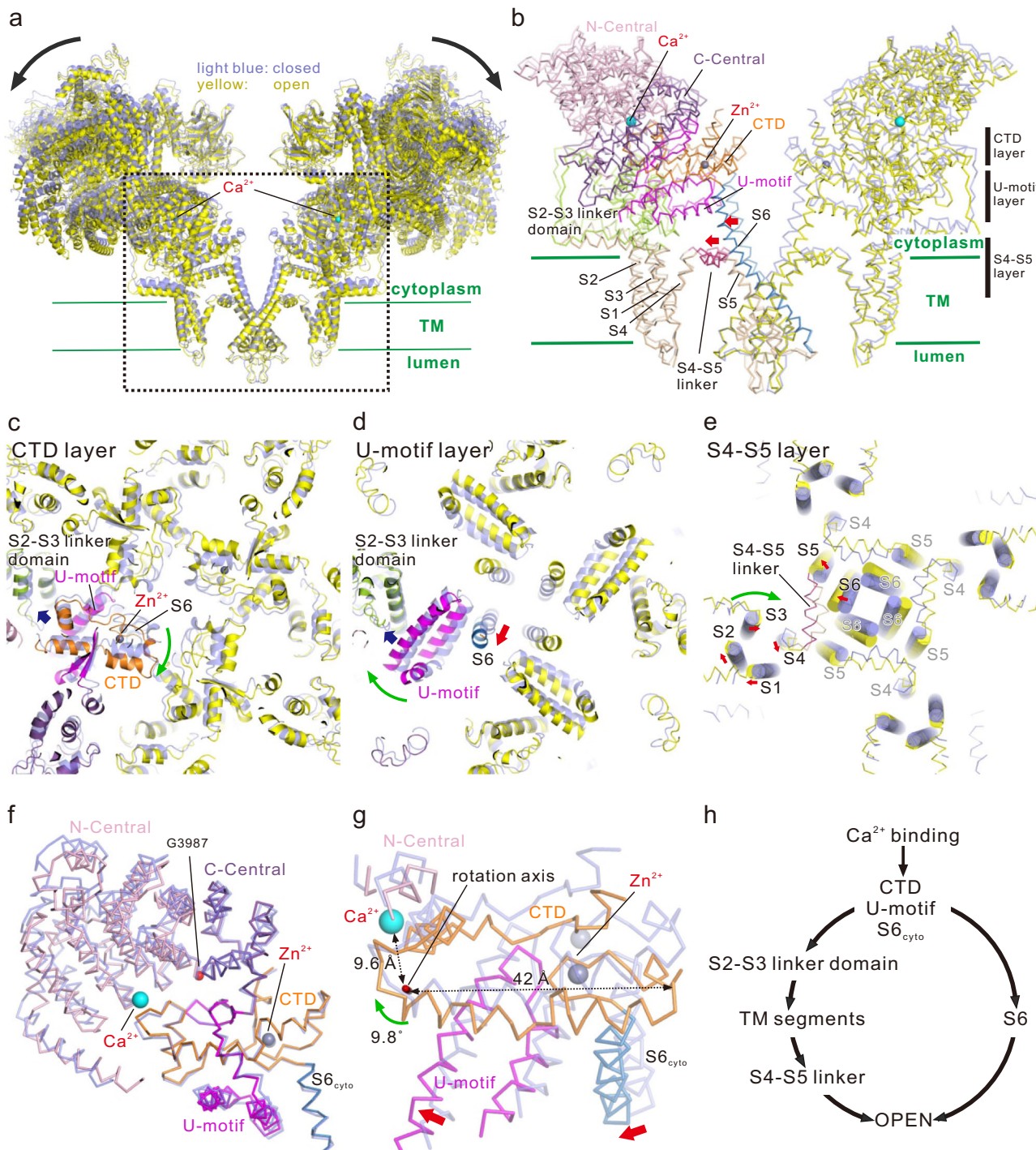

**Fig. 1 Conformational changes upon Ca²⁺ binding. a** Overlay of RyR2 in the closed (light blue) and open (yellow) states viewed from the direction parallel to the lipid bilayer is shown as a ribbon model. Two facing protomers in the RyR2 tetramer are shown. **b** Magnified view of the dotted box in (**a**). In the left protomer, each domain is colored (N-Central, light pink; C-Central, purple; U-motif, magenta; S1–S5, wheat; S2–S3 linker domain, light green; S4–S5 linker, warm pink; S6, blue; CTD, orange). S4–S5 linker and S6 moved outside upon Ca²⁺ binding as indicated by the red arrows. Three regions parallel to the membrane are defined as CTD, U-motif, and S4–S5 layers. Ca²⁺, shown as a cyan ball; Zn²⁺, shown as a gray ball. **c–e** Cross-section views of CTD, U-motif, and S4–S5 layers. The closed state is colored in light blue and the open state is colored according to (**b**) or yellow. In (**e**), Cα representation overlaid with cylindrical TM helices is used. Ca²⁺ binding causes clockwise rotation of CTD (green arrow in (**c**)), U-motif (green arrow in (**d**)), and S1-S4 TM helices and outward movement of S4–S5 linker and S6 (red arrows in (**e**)). **f** C-Central/U-motif/S6cyto/CTD complex. Closed (light blue) and open (colored according to (**b**)) states are overlaid at the CTD. Central domain is split into two parts at G3987 which works as the pivot of the rotation upon Ca²⁺ binding. **g** Rotation of the C-Central/U-motif/S6cyto/CTD complex upon Ca²⁺ binding viewed from the rotation axis. **h** Scheme of channel opening upon Ca²⁺ binding. Two independent pathways via S6 and S4–S5 linker are hypothesized. Source data are provided as a Source Data file.

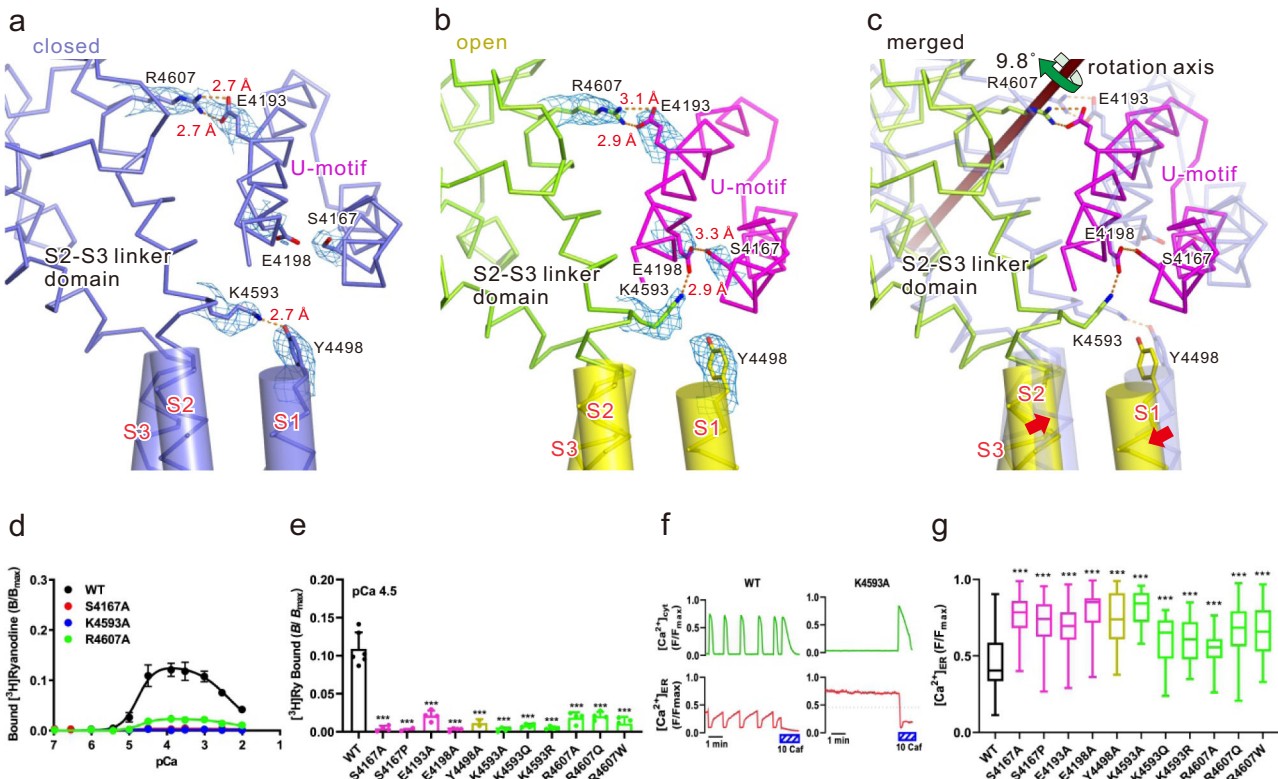

**Fig. 2 Key interactions between U-motif and S2-S3 linker domain upon channel opening. a–c** Interface of U-motif and S2-S3 linker domain in the closed state (**a**), open state (**b**), and an overlay of both states (**c**) viewed parallel to the membrane is shown as a Cα model. Amino acid residues involved in key interactions are shown as stick models. The color of carbon atoms is the same as that of Cα; oxygen, red; nitrogen, blue. The TM region forming α-helices is overlaid with the cylinder model. Hydrogen bonds or salt bridges are shown as orange dotted lines. Density maps around side chains shown in (**a**) and (**b**) are superimposed and contoured at 0.025. **d–g** Functional analysis of mutants involved in U-motif/S2–S3 linker domain interaction. **d** $Ca^{2+}$-dependent [$^3$H]ryanodine binding of WT and representative mutants. Data are shown as means ± SD ($n = 4$). **e** Summary of [$^3$H]ryanodine binding of WT and mutants at pCa 4.5. Data are shown as means ± SD ($n = 6$ and 4 for WT and mutants, respectively) and were analyzed by one-way ANOVA with Dunnett's multiple comparisons test. ***$p < 0.001$ from WT. **f** Representative traces of cytoplasmic ([$Ca^{2+}$]$_{cyt}$) and ER ([$Ca^{2+}$]$_{ER}$) $Ca^{2+}$ signals of HEK293 cells expressing WT or K4593A. Spontaneous $Ca^{2+}$ oscillations occurred with a concomitant decrease in [$Ca^{2+}$]$_{ER}$ in WT, while the K4593 mutant showed no $Ca^{2+}$ oscillations with an increased [$Ca^{2+}$]$_{ER}$, indicating loss-of-function of the channel. **g** Summary of the upper level of [$Ca^{2+}$]$_{ER}$ signals in WT and mutants. All mutants showed loss-of-function of the channel. Data are shown as box-and-whisker plots, with the median for all subjects shown as the center line, the box representing the 25–75 percentile, and the lines showing the range of the data ($n = 97, 68, 76, 73, 46, 35, 44, 36, 46, 54, 63,$ and 53 for WT, S4167A, S4167P, E4193A, E4198A, Y4498A, K4593A, K4593Q, K4593R, R4607A, R4607Q, and R4607W, respectively). Data were analyzed by one-way ANOVA with Dunnett's multiple comparisons test. ***$p < 0.001$ from WT. Source data are provided as a Source Data file.

**U-motif/S2–S3 linker domain interaction is a key for signal transduction in response to $Ca^{2+}$ binding.** In our hypothesis, the U-motif/S2-S3 linker domain interaction is important in transducing the rotation of the U-motif to TM segment movement (Fig. 1d, h and Supplementary movie 1, 2). We found three key interactions between the U-motif and S2–S3 linker domain formed by hydrogen bonds or salt bridges (Fig. 2a–c, Supplementary Fig. 3a, b and Supplementary movie 3). E4198-K4593-S4167 was evident in the open state, whereas Y4498-K4593 was only formed in the closed state, indicating that K4593 switches the interacting partner between the two states. Additionally, E4193-R4607 was stable in both states.

To clarify the roles of hydrogen bonds/salt bridges in channel gating, functional assays were conducted with recombinant RyR2 carrying mutations at the specific residues (Supplementary Table 2). $Ca^{2+}$-dependent [$^3$H]ryanodine binding was conducted as it reflects channel activity[9,33]. Wild-type (WT) RyR2 exhibited biphasic $Ca^{2+}$-dependent [$^3$H]ryanodine binding (Fig. 2d). This is explained by two independent $Ca^{2+}$ binding sites; binding of $Ca^{2+}$ to high-affinity site formed by N-Central and CTD opens the channel, whereas that to undetermined low-affinity site closes the channel[27,34]. Alanine substitution at K4593 markedly reduced

the binding (Fig. 2d). Two pathogenic mutations, K4593Q and K4593R[12], also led to the loss of binding (Fig. 2e and Supplementary Fig. 3c). In K4593R, two nitrogen atoms (N η1 and N η2) at the tip of Arg might occupy both oxygen atoms (O ε1 and O ε2) from E4198 in the open state, which prevents S4167-E4198 interaction essential for the channel opening. Similarly, binding was severely reduced after alanine substitutions (Y4498A, S4167A, and E4198A) and pathogenic mutation (S4167P)[12] in the interacting pairs (Fig. 2d, e and Supplementary Fig. 3c). The behaviors of mutant RyR2 channels were also evaluated in live cells by monitoring cytoplasmic and endoplasmic reticulum (ER) luminal $Ca^{2+}$ homeostasis[9,35]. In WT RyR2-expressing HEK293 cells, spontaneous $Ca^{2+}$ oscillations occurred with a concomitant decrease in [$Ca^{2+}$]$_{ER}$, indicating $Ca^{2+}$ release from the ER via RyR2 channels (Fig. 2f, left). In contrast, RyR2 channels carrying the K4593A mutation showed no such $Ca^{2+}$ oscillations with increased [$Ca^{2+}$]$_{ER}$ (Fig. 2f, right). 10 mM caffeine, a potent RyR activator, released $Ca^{2+}$ from ER in cells expressing K4593A, indicating that it forms a functional channel. These findings confirm a loss-of-function of the K4593A channel. Similar results were obtained with other substitutions in S4167, E4198, and K4593 (Fig. 2g, Supplementary Table 2). Altogether, these results

suggest that both E4198-K4593-S4167 and Y4498-K4593 interactions are important for channel opening. We also evaluated the E4193-R4607 interaction and found that alanine substitutions (E4193A and R4607A) and pathogenic mutations (R4607Q and R4607W) led to a loss-of-function of the channel (Fig. 2d–g and Supplementary Fig. 3c). These residues and surrounding sequences are well conserved in all three mammalian RyR subtypes (Supplementary Fig. 3d). These findings indicate that three interactions above S2 in the U-motif/S2-S3 linker domain interface play a critical role in transducing the $Ca^{2+}$-binding signal to S2 and that loss of these interactions results in a loss of channel function.

**Movements of the S1–S4 bundle lead to outward movement of the S4–S5 linker**. We hypothesize that movement of the S2–S3 linker domain is an important step in the channel opening through the coordinated movement of transmembrane segments (Fig. 1d, e, h). Movement of the S2-S3 linker domain causes S2 movement (Fig. 2c and Supplementary movie 3). This leads to the coordinated 7.6° clockwise rotation of S1, S3, and S4 (Figs. 1e, 3a–c, Supplementary Fig. 4a–d and Supplementary movie 1, 4). S1, S2, S3, and S4 are arranged in a circle, placed at equal intervals in a clockwise direction (Fig. 3a–c and Supplementary Fig. 4a–d). We found interactions between S1 and S2 (S1/S2), S2/S3, S3/S4, and S1/S4, all of which were maintained in both closed and open states (Fig. 3a, b, Supplementary Fig. 4a–d and Supplementary movie 4). The hydrophobic interactions between F4497 and L4592 at S1/S2; the hydrogen bond between Y4589 and D4715 at S2/S3; hydrogen bond between Y4720 and D4744 at S3/S4; and the salt bridge between R4501 and D4744 at S1/S4 appear to bundle the four TM segments into one (i.e., S1–S4 bundle).

Alanine substitution of residues involved in S1/S2 (F4497 and L4592) and S2/S3 (Y4589 and D4715) interactions exhibited significantly reduced [3H]ryanodine binding (Fig. 3d, e and Supplementary Fig. 5a), loss of $Ca^{2+}$ oscillations, and increased ER $Ca^{2+}$ (Fig. 3f, g)—all indicative of loss-of-function. The pathogenic mutant F4497C also resulted in loss-of-function (Fig. 3e, g and Supplementary Fig. 5a). Thus, S1/S2 and S2/S3 interactions are necessary for channel opening. In contrast, alanine substitution or pathogenic mutations of residues involved in S1/S4 (R4501 and D4744) and S3/S4 (Y4720 and D4744) interactions caused gain-of-function of the channel with increased [3H]ryanodine binding and reduction in ER $Ca^{2+}$ (Fig. 3d-g and Supplementary Fig. 5a). Especially, D4744A, which is involved in both interactions, exhibited greatly increased [3H]ryanodine binding with an enhanced $Ca^{2+}$ sensitivity for activation and loss of $Ca^{2+}$ inactivation (Fig. 3d and Supplementary Fig. 5c, d). These residues are well conserved in all the mammalian RyR subtypes (Supplementary Fig. 5e).

In our hypothesis, rotation of the S1–S4 bundle causes movement of the S4–S5 linker to open the channel (Fig. 1e, h). We found that the upper part of S4 rotated 40° counterclockwise to form an α-helix upon channel opening, which dramatically changed the position of F4749 (Fig. 3h–j, Supplementary Fig. 4e–g and Supplementary movie 4). The S4–S5 linker, which is unfolded and significantly bent in the direction of S6 in the closed state, rewinds to an α-helix and moves outward (Supplementary Fig. 4e–h and Supplementary movie 4). Hydrophobic interaction between F4749 in the upper part of S4 and L4505 in S1 was identified in the closed state (Fig. 3h and Supplementary Fig. 4a, h).

Alanine substitutions of residues involved in hydrophobic interactions (L4505A and F4749A) and the pathogenic mutant L4505P caused gain-of-function with increased $Ca^{2+}$ sensitivity and loss of $Ca^{2+}$ inactivation (Fig. 3k–n and Supplementary

Fig. 5b–d). Notably, no additive effects were observed upon binding of [3H]ryanodine to double mutant L4505A_F4749A (Supplementary Fig. 5b), supporting an interaction between the two residues. Whereas L4505 is conserved in all the mammalian RyR subtypes, F4749 is mutated to valine in the RyR1 subtype (Supplementary Fig. 5e). However, unlike the alanine mutant, F4749V exhibited biphasic $Ca^{2+}$ dependent [3H]ryanodine binding with a slightly reduced peak value (Supplementary Fig. 5b).

Taking these findings into consideration, we hypothesized a molecular mechanism in which a series of movements regulates channel opening. A brief summary of the roles of the key residues located on the S1–S4 bundle is shown as a scheme (Fig. 3o). In the absence of $Ca^{2+}$, the channel is stabilized in the closed state as the S4-S5 linker is unfolded and bent in the direction of S6 by the "stopper", in the form of L4505-F4749 interaction. The L4505-F4749 interaction prevents the α-helix formation of the upper part of S4 and is supported by S1/S4 and S3/S4 interactions that keep the two residues appropriately placed. Upon $Ca^{2+}$ binding, clockwise rotation of the S1–S4 bundle induced by U-motif/S2-S3 linker domain interaction alters the relative positions of L4505 and F4749 to release the stopper. This allows the α-helix formation of the upper part of S4 and subsequent α-helix formation and outward movement of the S4–S5 linker to open the channel. Since the α-helix formation of the S4–S5 linker shortened its length in the open state (Fig. 3o, Supplementary Fig. 4g), we assume that the rewinding to α-helix in the upper part of S4 provides a margin for shortening of the S4-S5 linker. Loss of necessary interactions to release the stopper, i.e., U-motif/S2-S3 linker domain, S1/S2, and S2/S3, leads to loss-of-function. Conversely, loss of interactions that support S4, i.e., S1/S4 and S3/S4, spontaneously releases the stopper to cause a gain-of-function. This well explains the differential roles of TM helices on the channel gating (Fig. 3o).

**U-motif plays a key role in stabilizing the channel in the closed state**. U-motif is a key component for the RyR2 channel opening as described above (Fig. 1d, h). It constitutes the U-motif/S6cyto/CTD complex, which was stable in both the closed and open states (Fig. 1f). In the closed state, we found a close contact between U-motif and S6cyto (Fig. 4a, b Supplementary Fig. 6a and Supplementary movie 5), which was similar as observed in the closed state structure of RyR1[36]. In the closed state, F4173, V4176, and N4177 in the U-motif and Q4875 and V4879 in S6cyto faced each other forming van der Waals interactions, and were surrounded by Q4876 and Q4878 from S6 (Fig. 4b, Supplementary Fig. 6a and Supplementary movie 5). In the open state, S6cyto self-rotated (~30° clockwise at Q4875) and U-motif/S6cyto loosened and appeared unstable (Fig. 4c, Supplementary Fig. 6a–c and Supplementary movie 5). Buried surface area analysis calculated by CNS[37] demonstrated that the U-motif/S6cyto interface in the open state (362 $Å^2$) was smaller than in the closed state (514 $Å^2$). The U-motif also interacted with CTD via F4888, where it penetrated partly into the hydrophobic pocket formed by U-motif F4171, I4172, and V4175 and CTD L4914 (Fig. 4d, Supplementary Fig. 6d and Supplementary movie 5). The U-motif/CTD interaction was stable in both states.

We also noticed compaction of U-motif upon channel opening, which was induced by a 2.5 Å parallel shift of its N-terminal helix (4161–4178) toward the C-terminal side (4183–4205) (Fig. 4e–g). Upon binding of $Ca^{2+}$ to CTD, the upper part of S6cyto rotated together with CTD (Figs. 1g, 4g). Along with the rotation, S6cyto pushed the N-terminal helix of U-motif, thereby, the parallel shift of the N-terminal helix occurred. This parallel shift seems to be the key trigger for releasing the restraint of S6cyto from U-motif,

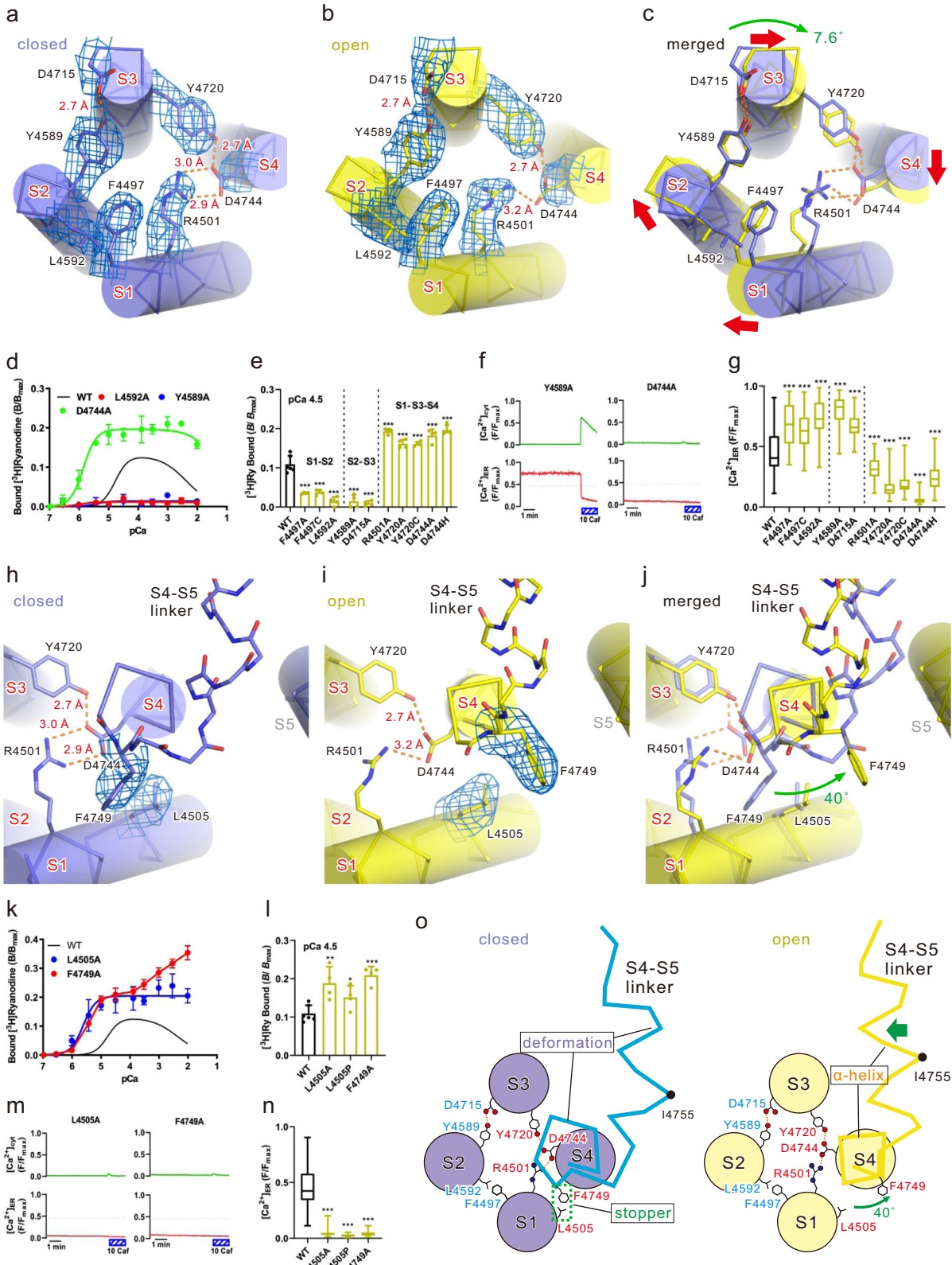

allowing further rotation and outward movement of S6$_{cyto}$ (Fig. 4c, g and Supplementary Fig. 6a). Two hydrogen bonds (S4167-E4198 and E4198-K4593) newly formed upon opening seem to be the key for stabilizing this shift. Other important factors seem to be π–π interaction formed by F4171-F4191 and two consecutive glycines (G4179 and G4180) located on the loop connecting the N-terminal and the C-terminal helix of the

U-motif (Fig. 4g and Supplementary Fig. 6e). The former may function as a guide for the shift of the N-terminal helix (Fig. 4g, inset), and the latter may function as the flexible hinge to enable the deformation of the U-motif.

All alanine-substituted mutants of residues interacting between U-motif and S6$_{cyto}$ as well as two CPVT mutations (N4177S and N4177Y) led to gain-of-function (Fig. 4h–k) with mild enhancement

**Fig. 3 Key interactions in the transmembrane region upon channel opening. a–c** The S1–S4 bundle. Closed state (**a**), open state (**b**), and overlay of the structures in both states (**c**) are shown as a Cα model and overlaid with cylinder models. Hydrogen bonds/salt bridges are shown as orange dotted lines. Density maps around side chains shown in (**a**) and (**b**) are superimposed and contoured at 0.03. **d–g** Functional analysis of mutants involved in the S1/S2, S2/S3, S1/S4, or S3/S4 interaction. **d** $Ca^{2+}$-dependent $[^3H]$ryanodine binding of WT and representative mutants. Data are shown as means ± SD ($n = 4$). **e** Summary of $[^3H]$ryanodine binding of WT and mutants at pCa 4.5. Data are shown as means ± SD ($n = 6$ and 4 for WT and mutants, respectively) and were analyzed by one-way ANOVA with Dunnett's test. ***$p < 0.001$ from WT. **f** Representative traces of $[Ca^{2+}]_{cyt}$ and $[Ca^{2+}]_{ER}$ signals of HEK293 cells expressing Y4589A or D4744A. **g** Summary of the upper level of $[Ca^{2+}]_{ER}$ signals in WT and mutants. Data are shown as box-and-whisker plots, with the median for all subjects shown as the center line, the box representing the 25–75 percentile, and the lines showing the range of the data ($n = 97, 69, 65, 66, 35, 56, 74, 54, 63, 39,$ and 94 for WT, F4497A, F4497C, L4592A, Y4589A, D4715A, R4501A, Y4720A, Y4720C, D4744A, and D4744H, respectively) and were analyzed by one-way ANOVA with Dunnett's test. ***$p < 0.001$ from WT. **h–j** TM region around S4. Closed state (**h**), open state (**i**), and overlay of the structures in both states (**j**). Structures in both states were fitted to the bottom part of S4 helices and viewed along the axis of the S4 helix and from the cytoplasm; the Cα model overlaid with cylinder models. Main chain representation of the S4-S5 linker. Density maps around side chains shown in (**h**) and (**i**) are superimposed and contoured at 0.03. **k** $Ca^{2+}$-dependent $[^3H]$ryanodine binding of WT and representative mutants. Data are shown as means ± SD ($n = 4$). **l** Summary of $[^3H]$ryanodine binding of WT and mutants at pCa 4.5. Data are shown as means ± SD ($n = 6$ and 4 for WT and mutants, respectively) and were analyzed by one-way ANOVA with Dunnett's test. *$p < 0.05$, **$p < 0.01$, ***$p < 0.001$ from WT. **m** Representative traces of $[Ca^{2+}]_{cyt}$ and $[Ca^{2+}]_{ER}$ signals of HEK293 cells expressing L4505A or F4749A. **n** Summary of the upper level of $[Ca^{2+}]_{ER}$ signals in WT and mutants. Data are shown as box-and-whisker plots, with the median for all subjects shown as the center line, the box representing the 25–75 percentile, and the lines showing the range of the data ($n = 97, 67, 35,$ and 74 for WT, L4505A, L4505P, and F4749A, respectively) and were analyzed by one-way ANOVA with Dunnett's test. ***$p < 0.001$ from WT. **o** Scheme of the structure in the S1-S4 TM helices and S4-S5 linker. S1-S4 helices are drawn with a circle, and the positions of Cα atoms of S4-S5 linker are connected by a line. Amino-acid numbers associated with signal transduction and stabilization are shown in blue and red, respectively. While the S1/S2 and S2/S3 interactions are involved in signal transduction, the S1/S4 and S3/S4 interactions are involved in the stabilization of the channel in the closed state. Alteration of the relative positions of L4505 and F4749 to release the stopper shown as the green dotted box. This allows α-helix formation of the upper part of S4 and subsequent α-helix formation and outward movement of the S4–S5 linker to open the channel. Since α-helix formation of the S4–S5 linker shortened its length in the open state, the rewinding to α-helix in the upper part of S4 provides a margin for shortening of the S4–S5 linker. The position of the Cα atom of I4755 are shown as a black-filled circle. Source data are provided as a Source Data file.

in $Ca^{2+}$ sensitivity and severe loss of $Ca^{2+}$ inactivation (Supplementary Fig. 7a, c, d). Alanine substitution and pathogenic mutation of F4888 exhibited severe gain-of-function with increased $Ca^{2+}$ sensitivity and loss of $Ca^{2+}$ inactivation (Fig. 4l–o and Supplementary Fig. 7b–d). Alanine substitution of F4171, I4172, V4175, and L4914—involved in the hydrophobic pocket—also led to milder gain-of-function of the channel than F4888 mutants (Fig. 4l–o and Supplementary Fig. 7b–d). Notably, no additive effects were observed with N4177A and F4888A, indicating that U-motif/S6$_{cyto}$ and U-motif/CTD interactions are involved in the common pathways (Supplementary Fig. 7b). All the residues thought to be involved in the U-motif/S6cyto/CTD interactions and surrounding sequences are conserved in mammalian three RyR subtypes (Supplementary Fig. 7e). These findings suggest that interactions within the U-motif/S6$_{cyto}$/CTD complex play a key role in stabilizing the closed state.

Alanine-substitutions of F4171 and F4191 showed gain-of-function of the channel (Fig. 4m, o and Supplementary Fig. 7b–d). No additive effects were observed with double mutant, F4171A_F4191A (Supplementary Fig. 7b), supporting π–π interaction of the two residues. The gain-of-function, however, was opposite to the effect of alanine substitution of S4167, E4198, and K4593, residues involving the compaction of U-motif (Fig. 2 and Supplementary Fig. 4e–g). This can be explained as follows. In WT, the aromatic rings of F4171 and F4191 form a stable π–π interaction, behaving like a spring that suppresses the compaction of the U-motif (Supplementary Fig. 6f). Alanine substitution of F4171 or F4191 weakens the π–π interaction to break the spring, which accelerates the compaction of U–motif.

**Structural basis of the loss-of-function mutation.** The structure-based functional analysis described above validated our hypothesis of the gating mechanism of the RyR2 channel (Fig. 1h). To further prove the hypothesis, we conducted a structural analysis of a loss-of-function mutant (K4593A) (Fig. 2d–g). K4593 is the key residue for U-motif-S2-S3 linker domain interaction (Fig. 2a–c) and compaction of U-motif (Fig. 4e–g). We expected that substitution of this residue with

alanine may prevent transduction of the $Ca^{2+}$-binding signal from U-motif to the S2-S3 linker domain. We have determined the structures of K4593A mutant in the presence of 1 mM EGTA (K4593A(EGTA)) at 3.3 Å resolution (Supplementary Fig. 8a) and in the presence of 100 μM $Ca^{2+}$ (K4593A($Ca^{2+}$)) at 3.8 Å resolution (Supplementary Fig. 8b). The density for the side chain of K4593 completely disappeared in both structures (Supplementary Fig. 8c). The density corresponding to $Ca^{2+}$ was clearly visible at the $Ca^{2+}$ binding site only in the structure of K4593A($Ca^{2+}$) (Supplementary Fig. 8d). K4593A(EGTA) showed a closed state which is very similar to that of WT (Fig. 5a and Supplementary Fig. 9a, c). In contrast, K4593A($Ca^{2+}$) exhibited large conformational changes around the cytoplasmic region like the open state of WT but essentially no change in the TM region (Fig. 5b and Supplementary Fig. 9b, d).

Changes associated with $Ca^{2+}$ binding were analyzed at three layers (Fig. 5c–e, Supplementary Fig. 9c, d and Supplementary movie 6). Upon $Ca^{2+}$ binding, CTD rotated clockwise (4.2°, Fig. 5c). In contrast, U-motif and the TM helices hardly moved (Fig. 5d, e). Although a slight movement was observed in the upper part of the S2-S3 linker domain in the structure of K4593A($Ca^{2+}$) compared to the structure in K4593A(EGTA) (Fig. 5f), essentially no movement was detected juxtamembrane region of the S1–S4 bundle (Fig. 5f). The rotation of CTD in K4593($Ca^{2+}$) (4.2°) is about half of that in WT in the open state (9.8°, Fig. 5g). No compaction of U-motif was detected in K4593A($Ca^{2+}$) (Fig. 5h), indicating that the hydrogen bond between K4593 and E4198 is indispensable for the compaction of U-motif. The subsequent outward movement of S6$_{cyto}$ was not detected either (Fig. 5h). On the other hand, no considerable difference was detected between K4593A(EGTA) and WT in the closed state at three layers (Supplementary Fig. 9c) nor the shape of U-motif (Supplementary Fig. 9e), supporting that the above changes detected in K4593A($Ca^{2+}$) was derived from the effect of K4593A mutation. These lines of evidence support our hypothesis that compaction and rotation of U-motif move the S2–S3 linker domain to open the channel pore.

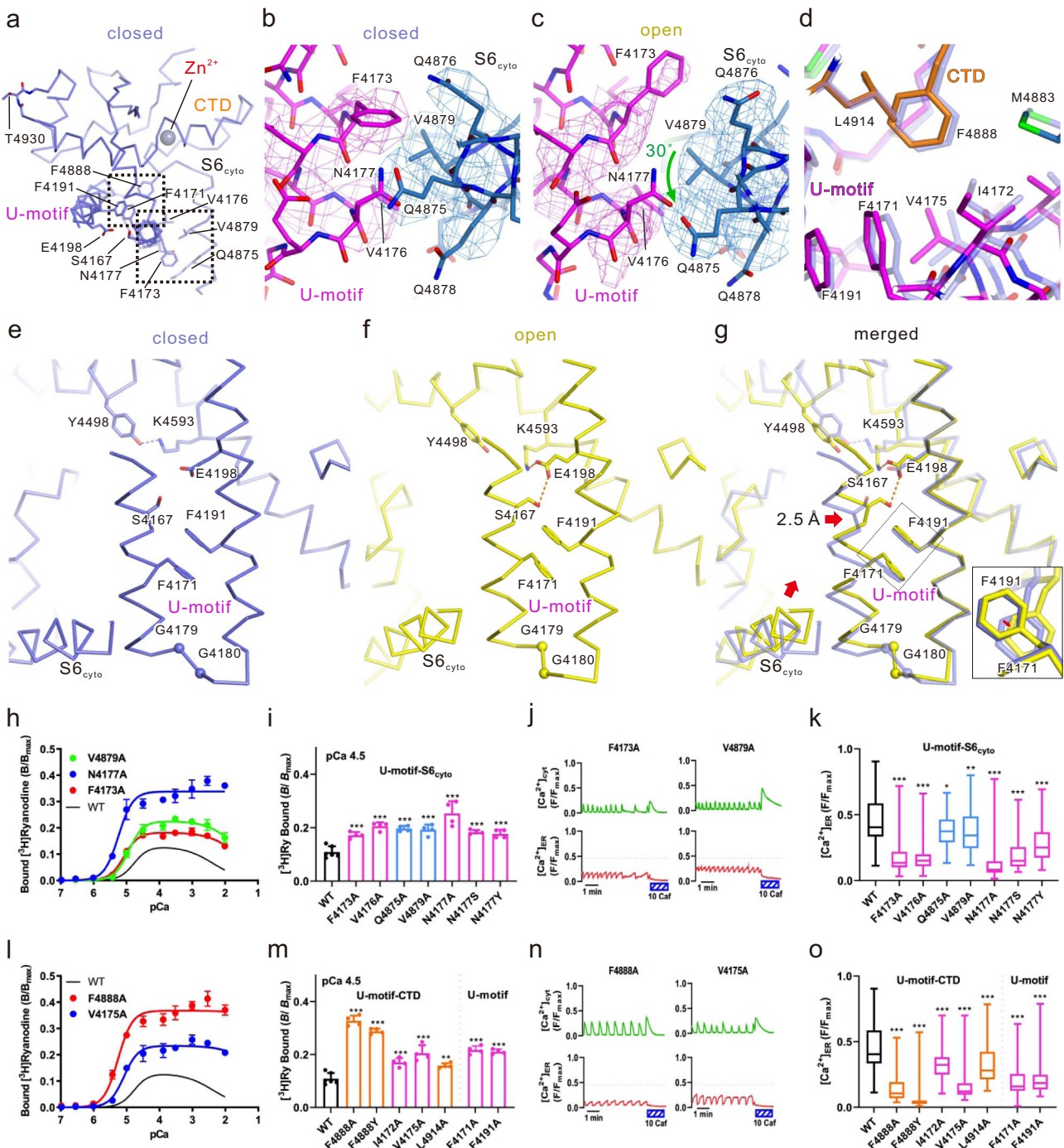

It is interesting that the rotation due to Ca²⁺ binding occurred only in CTD and the upper part of S6$_{cyto}$ (Fig. 5c, g, Supplementary Fig. 9f). Since CTD is placed between 1-turn β sheet connected just before U-motif (Supplementary Fig. 9f), U-motif, CTD, and S6$_{cyto}$ are supposed to rotate together upon Ca²⁺ binding as observed in the open state (Fig. 1f, Fig. 5c, g, Supplementary Fig. 9f). The independent movement of CTD in K4593A indicates that CTD placed between 1-turn β sheet and U-motif have some degree of freedom in the rotation. The upper part of S6$_{cyto}$ rotated together with CTD because of a direct connection to CTD (Supplementary Fig. 9f). However, the rotation throughout S6 was blocked by the collision with the uncompacted U-motif in K4593A (Supplementary Fig. 9f).

**Summary of the Ca²⁺ induced movements in RyR2.** Here, we summarize the channel opening mechanism associated with Ca²⁺ binding of RyR2 determined by high-resolution cryo-EM structures: (1) binding of Ca²⁺ rotates the U-motif/S6$_{cyto}$/CTD complex by 9.8° along with the axis which is 40 Å apart from S6$_{cyto}$; (2) compaction of U-motif; (3) extruding the S2–S3 linker domain by the U-motif causes S2 to move; (4) movement of S2 rotates the S1–S4 bundle by 7.6° to release the stopper comprising F4749/L4505; (5) rotation of the upper part of S4 allows outward movement of the S4–S5 linker, creating a space where S6 can lean into; and (6) rotation of S6$_{cyto}$ by 30° causes the outward leaning of S6 by 7.8 Å (Fig. 5i). Thus, the sequential rotations of domains/α-helices and domain-domain interactions may occur in conformation changes associated with Ca²⁺ binding. Rotation of

**Fig. 4 Key interactions between the U-motif and S6$_{cyto}$/CTD. a** Structure around the U-motif in the open state (U-motif, magenta; S6$_{cyto}$, blue; CTD, orange). Ca$^{2+}$ and Zn$^{2+}$ are shown as cyan and gray spheres, respectively. **b, c** Details of the U-motif/S6$_{cyto}$ interaction. Structures in the closed (**b**) and open (**c**) states fitted to the N-terminal region of the U-motif are shown as a full atomic model. Density maps around the interaction are superimposed and contoured at 0.025. **d** Details of the U-motif/CTD interaction around F4888. Overlay of the structures in the closed (light blue) and open (colored) states. **e–g** Compaction in U-motif. Closed state (**e**), open state (**f**), and overlay of the structures in both states (**g**) are shown as a Cα model. Inset of (**g**) shows the magnified view of the F4171/F4191 stacking looking from aromatic ring of F4191. F4171 is stacked in parallel with F4191 in both the closed and the open states, but it slightly moves away from F4191 in the open state. The structures are fitted in the C-terminal side of U-motif (4183–4205). The closed state and the open state is colored with light blue and yellow, respectively. The N-terminus side of the U-motif of the WT in the open state is ~2.5 Å closer to the C-terminus side of the U-motif as indicated by the red arrow, and as a result, S6$_{cyto}$ movement (red arrow) becomes possible. **h–o** Functional analysis of mutants involved in U-motif/S6$_{cyto}$ (**h–k**) and U-motif/CTD (**l–o**) interactions. **h, l** Ca$^{2+}$-dependent [$^3$H]ryanodine binding of WT and representative mutants. Data are shown as means ± SD ($n = 4$). **i, m** Summary of [$^3$H]ryanodine binding of WT and mutants at pCa 4.5. Data are shown as means ± SD ($n = 6$ and 4 for WT and mutants, respectively) and were analyzed by one-way ANOVA with Dunnett's test. **$p < 0.01$, ***$p < 0.001$ from WT.
**j, n** Representative traces of [Ca$^{2+}$]$_{cyt}$ and [Ca$^{2+}$]$_{ER}$ signals of HEK293 cells expressing F4173A or V4879A (**j**) and F4888A or V4175A (**n**). **k, o** Summary of the upper level of [Ca$^{2+}$]$_{ER}$ signals in WT and mutants. Data are shown as box-and-whisker plots, with the median for all subjects shown as the center line, the box representing the 25–75 percentile, and the lines showing the range of the data ($n = 97$, 96, 97, 89, 87, 96, 54, and 49 for WT, F4173A, V4176A, Q4875A, V4879A, N4177A, N4177S, and N4177Y, respectively, for (**k**) and 97, 34, 59, 58, 98, 63, 94, and 94 for WT, F4888A, F4888Y, I4172A, V4175A, L4914A, F4171A, and F4191A, respectively, for (**o**)) and were analyzed by one-way ANOVA with Dunnett's test. *$p < 0.05$, **$p < 0.01$, ***$p < 0.001$ from WT. Source data are provided as a Source Data file.

U-motif in K4593A in the presence of Ca$^{2+}$ stopped in the middle of the transition from the closed state to the open state, that is, binding of Ca$^{2+}$ rotates only U-motif by 4.2° along with the same axis as the rotation from closed state to open state (Fig. 5j).

## Discussion
By combining high-resolution cryo-EM structures and quantitative functional analysis of mutant channels, we successfully clarified the gating mechanism of RyR2 upon Ca$^{2+}$ binding. Figure 6a shows a schematic diagram of the channel core domain of RyR2 with details of the important interactions identified in this study. Mutations of residues involved in the signaling pathway for outward movement of the S4-S5 linker caused loss-of-function, whereas those supporting the stopper (i.e., S1/S3, S3/S4, and S1/S4 interactions) or the U-motif/CTD/S6$_{cyto}$ complex led to gain-of-function. Given the locations of these mutations, it becomes apparent that interactions close to the channel pore are important for stabilizing the channel in the closed state and those in the surrounding region are essential for channel opening.

The present results can reasonably explain the gating mechanism of RyR2 channel and how pathogenic or alanine mutations affect the channel activity (Fig. 6b–d). In the closed state of the WT channel, the S4–S5 linker prevents outward leaning of S6 (Fig. 6b, left). In the open state of the WT, binding of Ca$^{2+}$ moves CTD, which transmits signals into two downstream pathways to open the channel; S4-S5 linker is unlocked by sequential conformational changes beginning at the U-motif and S6 leans outward by the movement of S6$_{cyto}$ (Fig. 6b, right). In addition to direct regulation of the channel gate, S6$_{cyto}$ interacts with U-motif to restrict its movement toward the S2–S3 linker domain (blue T-shaped line, Fig. 6b, left). This negative regulation by S6$_{cyto}$ is supported by hydrophobic interaction between CTD (F4888) and U-motif (dotted line, Fig. 6b, left). Since the interaction is still maintained at the open state, the WT channel is not fully activated (Fig. 6b, right). Mutations in the U-motif/S2–S3 linker domain, S1/S2, or S2/S3 interface disrupt the interaction, which is essential for the channel opening (Fig. 6c, left). This interrupts Ca$^{2+}$-induced conformational changes, resulting in loss-of-function (Fig. 6c, right). Mutations in S3/S4 or S1/S4 interface unlock the S4–S5 linker (Fig. 6d, upper left). However, in the absence of Ca$^{2+}$, this does not induce spontaneous outward movement of the S4–S5 linker and the channel pore is kept closed. In the presence of Ca$^{2+}$, these channels will show gain-of-function since the S4–S5 linker is more mobile

(Fig. 6d, upper right). Mutations in the U-motif/S6$_{cyto}$ or the U-motif/CTD disrupt the interaction to make the U-motif and S6$_{cyto}$ more mobile (Fig. 6d, lower left). Upon Ca$^{2+}$ binding, these channels will show gain-of-function by increased mobility in the S4–S5 linker and S6$_{cyto}$ (Fig. 6d, lower right). Considering that residues involved in the interactions are highly conserved and pathogenic mutations in RyR1 (L4505P, D4744H, and F4888Y; see Supplementary Table 2) also showed corresponding behaviors, the fundamental mechanism of channel gating may be common among three RyR subtypes.

Many structures of pig RyR2 have been determined[24–26]. Among them, structures in the open state have been classified into two groups: PCB95 in addition to Ca$^{2+}$ (Ca$^{2+}$/PCB95)[24] and ATP/caffeine in addition to Ca$^{2+}$ (Ca$^{2+}$/ATP/caffeine)[26]. We determined open structure of RyR2 with Ca$^{2+}$ alone in high salt solution in which channel activity is greatly enhanced but Ca$^{2+}$-dependent activation is preserved (Supplementary Fig. 1c-e). We compared our structure in the open state with those structures by analyzing conformational changes upon Ca$^{2+}$ binding at the CTD layer and at the U-motif layer (Supplementary Fig. 10a). In the Ca$^{2+}$/PCB95 structure, rotation of CTD and U-motif occurred upon opening, thereby outward movement of S6 occurred (Supplementary Fig. 10a, top). In the Ca$^{2+}$/ATP/caffeine structure, in contrast, both CTD and U-motif made a translational outward movement toward caffeine (Supplementary Fig. 10a, middle). Conformational change in our structures was similar to that observed in the Ca$^{2+}$/PCB95 structure (Supplementary Fig. 10a, bottom). Ca$^{2+}$/PCB95 open structure was only determined in the absence of FKBP12.6 because the addition of FKBP12.6 closes the channel[26]. This is in contrast to our open structure which binds FKBP12.6 (Supplementary Fig. 1k).

Previous structures of RyR in the presence of Ca$^{2+}$ alone failed to get an open state; the channel pore was closed[19,26]. Instead, slight movements of CTD, U-motif, and S6 occurred upon Ca$^{2+}$ binding (referred to as 'primed' state). Analysis of conformational changes in CTD and U-motif layers revealed slight rotation of CTD and U-motif and slight outward movement of S6 in the primed state, indicating an intermediate state between closed and open states (Supplementary Fig. 10b, top). In contrast, our K4593A(Ca$^{2+}$) structure showed no movement of U-motif and S6 at all, reflecting disruption of transduction signal from Ca$^{2+}$ binding (Supplementary Fig. 10b, bottom). Thus, K4593A(Ca$^{2+}$) is a particular state caused by a specific mutation and is different from the reported primed state.

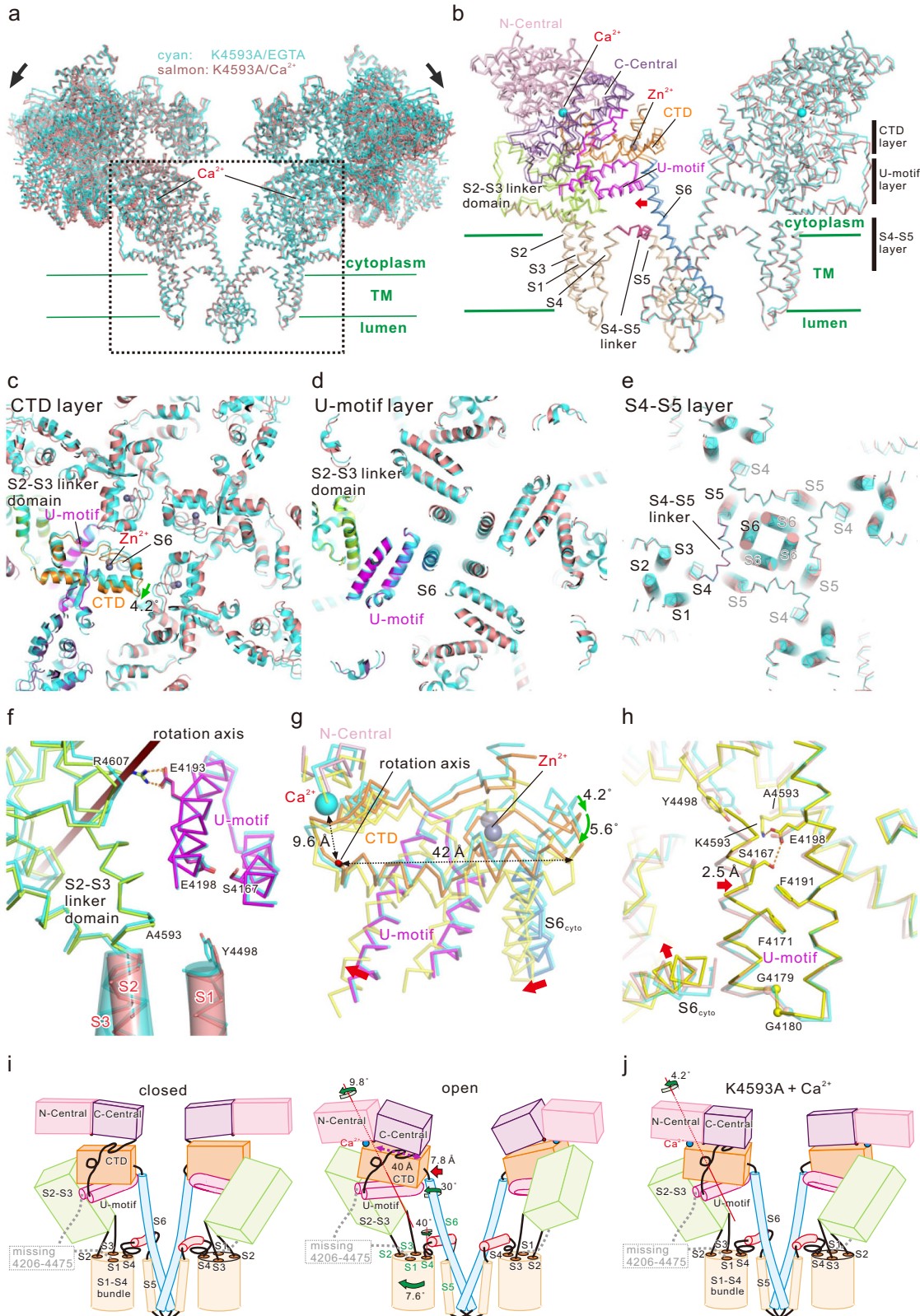

The outward movement of the S4–S5 linker is important in the channel opening, as it creates the space where S6 can lean into[19,24]. des Georges et al.[19] provided a more detailed description about the movement of the S4–S5 linker in RyR1; it is bent in the direction of S6 in the closed state, but it straightens and alters its contacts with S6, thereby opening the channel pore. We showed similar conformational changes in the S4–S5 linker in RyR2 and further demonstrated that rewinding of the upper part of S4 to an α-helix, which is caused by rotation of the S1–S4 bundle, drives the S4–S5 linker movement (Fig. 3h–j and Supplementary Fig. 4e–h). Regulatory mechanisms of interaction between the S4–S5 linker and S6 will be the next subject for a complete understanding of channel gating.

**Fig. 5 Structural basis of loss-of-function mutation and gating mechanism upon Ca$^{2+}$ binding. a** Overlay of K4593A mutant in the presence of EGTA (cyan) and K4593A mutant in the presence of Ca$^{2+}$ (salmon) viewed from the direction parallel to the lipid bilayer is shown as a ribbon model. Two facing protomers in the RyR2 tetramer are shown. **b** Magnified view of the dotted box in (**a**). In the left protomer, each domain is colored (N-Central, light pink; C-Central, purple; U-motif, magenta; S1–S5, wheat; S2–S3 linker domain, light green; S4–S5 linker, warm pink; S6, blue; CTD, orange). S6 moved outside upon Ca$^{2+}$ binding as indicated by the red arrow. Three regions parallel to the membrane are defined as CTD, U-motif, and S4–S5 layers. Ca$^{2+}$, shown as cyan ball; Zn$^{2+}$, shown as gray ball. **c–e** Cross-section views of CTD, U-motif, and S4–S5 layers. K4593A mutant in the presence of EGTA is colored with cyan and in the presence of Ca$^{2+}$ is colored according to (**b**) or salmon. In (**e**), Cα representation overlaid with cylindrical TM helices are used. Ca$^{2+}$ binding causes clockwise rotation of CTD (green arrow in (**c**)), but U-motif and S1-S4 TM helices show no considerable movement as shown in Figs. 1d, e. **f** An overlay of the interface of U-motif and S2-S3 linker domain in K4593A mutant in the presence of EGTA and Ca$^{2+}$ viewed parallel to the membrane and shown as a Cα model. Amino acid residues involved in key interactions are shown as stick models. The color of carbon atoms is the same as that of Cα; oxygen, red; nitrogen, blue. The TM region forming α-helices is overlaid with the cylinder model. Hydrogen bonds or salt bridges are shown as orange dotted lines. **g** Rotation of the C-Central/U-motif/S6$_{cyto}$/CTD complex upon Ca$^{2+}$ binding viewed from the rotation axis. Each domain of the K4593A mutant in the presence of Ca$^{2+}$ is colored (U-motif, magenta; S6, blue; CTD, orange). K4593A mutant in the presence of EGTA and WT in the open state is shown as cyan and yellow, respectively. **h** Overlay of three structures (K4593A mutant in the presence of EGTA, K4593A mutant in the presence of Ca$^{2+}$, and WT in the open state) fitted in the C-terminal side of U-motif (4183–4205). K4593A mutant in the presence of EGTA, K4593A mutant in the presence of Ca$^{2+}$, and WT in the open state is shown as cyan, salmon, and yellow, respectively. The N-terminus side of the U-motif of the WT in the open state is ~2.5 Å closer to the C-terminus side of the U-motif as indicated by the red arrow, and as a result, S6$_{cyto}$ movement (red arrow) becomes possible. This movement was not observed in the K4593A mutant in the presence of Ca$^{2+}$. **i, j** Scheme of the structure in the closed (**i**, left), open state (**i**, right) and K4593A mutant in the presence of Ca$^{2+}$ (**j**). **i** In the closed state, the upper part of S4 does not form an α-helix. The S4–S5 linker is unfolded and significantly bends in the direction of S6. In the open state, binding of Ca$^{2+}$ to the C-Central/CTD interface causes 9.8° rotation of all domains consisting of C-Central/U-motif/S6$_{cyto}$/CTD complex and compaction of U-motif, leading to two pathways. Pathway 1: the rotation causes 30° rotation of S6$_{cyto}$ which loosens the U-motif/S6$_{cyto}$ interaction and allows outward movement of S6. Pathway 2: a sequential movement of the S2–S3 linker domain, S2, S1–S4 bundle, and S4 allows the upper part of S4 to rewind and form an α-helix. Subsequently, the S4–S5 linker moves outward, creating a space where S6 can lean into. A combination of these two independent pathways eventually leads to the opening of the channel. **j** In the K4593A mutant, binding of Ca$^{2+}$ to the C-Central/CTD interface causes 4.2° rotation of the C-Central/U-motif complex, but no substantial movement occurs in the U-motif and S6$_{cyto}$, therefore, the signal of Ca$^{2+}$ binding is not transmitted to the subsequent steps, and the opening of the channel pore does not occur.

In RyR1, rearrangement of the salt bridges between S1 (R4563), S3 (Y4791) and S4 (D4815) has been reported to occur upon channel opening[19]. We showed that the corresponding salt bridges between R4501, Y4720, and D4744, which are critical to keeping S4 in place for appropriate positioning of the stopper (L4505-F4749 interaction), were maintained in both closed and open states in RyR2 (Fig. 3 and Supplementary Fig. 4). Interestingly, a recent study showed that diamide insecticides bind to the pocket within the S1–S4 bundle of RyR1, thereby disrupting the salt bridge to open the channel[38]. Our results reasonably explain why such insecticides activate the RyR channel.

The U-motif is proposed to be important in channel gating as it forms a stable complex with S6$_{cyto}$ and CTD, which move together when the channel opens[36]. This movement is accompanied by a rigid shift in the S2–S3 linker domain, which has been proposed to eventually open the channel pore[22]. We found that interaction of the U-motif with the S2–S3 linker domain is critically important in the channel opening; disruption of the interaction causes loss-of-function of the channel (Fig. 2 and Supplementary Fig. 3). In addition, we demonstrated that the U-motif/S6$_{cyto}$ interaction stabilizes the channel to the closed state (Fig. 4 and Supplementary Figs. 6, 7). Stabilization of the channel is important not only for the prevention of the hyperactivity but also for regulation by various stimuli or modifications that activate the channel, e.g., phosphorylation or oxidation/S-nitrosylation[2,13]. The U-motif/S6$_{cyto}$ interaction is a stabilization mechanism first identified at the molecular level and thus may contribute to the regulation of the RyR channel.

In RyRs, the large cytoplasmic region is believed to play an important role in regulating channel opening. Indeed, a number of pathogenic mutations are located in the cytoplasmic region. The question arises as to how these mutations, located far from the core domain, cause changes in the channel activity. We demonstrated that the C-Central domain forms a tight complex with the U-motif/S6$_{cyto}$/CTD to move together upon Ca$^{2+}$ binding (Fig. 1f and Supplementary Fig. 2e, f). The N-Central domain also moves upon Ca$^{2+}$ binding (Fig. 1f). Through the N-/

C-Central domains, the core domain has a series of interactions with the N-terminal domains (NTD), HANDLE, and HD1 domains which consist a major part of the large cytoplasmic region[24,36]. Thus, it is possible that the conformational changes of the cytoplasmic region by pathogenic mutations affect the channel gating through these interactions. Recently, RyR1 and RyR2 structures carrying gain-of-function mutations at N-terminal domains (NTD) were reported[28]. Determining the structures of various mutant RyRs will help to elucidate the long-range allosteric gating mechanism in the future.

In conclusion, we have revealed the gating mechanism of RyR2 upon Ca$^{2+}$ binding and provided structural insights into the effects of pathogenic mutations on channel activity. These findings may greatly help develop more effective drugs to treat RyR2-associated diseases.

## Methods

**Expression and purification of SBP-FKBP12.6.** Expression and purification of SBP-tagged FKBP12.6 were performed as described previously[30] with some modifications. Briefly, cDNA for SBP-tagged human FKBP12.6 was subcloned into the pET28 vector (Novagen) containing a 6×His-tag at the N-terminus. *Escherichia coli* BL21 cells (Novagen) transformed with the above expression vector were grown at 37 °C in 500 mL LB medium containing ampicillin at a final concentration of 50 μg/mL. After reaching an OD$_{600}$ of 0.95–1.2, SBP-FKBP12.6 expression was induced by adding 1 mM IPTG (Wako) for 3 h at 37 °C. Cells were then harvested, resuspended in 20 mM MOPS-Na (pH 7.4), 300 mM NaCl, 20 mM imidazole, and a cocktail of protease inhibitors [antipain (Peptide Institute), aprotinin (Nacalai), chymostatin (Peptide Institute), leupeptin (Peptide Institute) and pepstatin A (Peptide Institute), each 2 μg/mL], and lysed using a sonicator in an ice bath. The lysate was centrifuged at 100,000 × *g* for 30 min at 4 °C and the pellet was discarded. The supernatant was incubated for 1 h at 4 °C with 0.5 mL Profinity IMAC resin (Bio-Rad Laboratories, Hercules, CA) in a buffer containing 20 mM MOPS-Na (pH 7.4), 300 mM NaCl, and 20 mM imidazole. The resin was washed five times with the buffer and SBP-FKBP12.6 was eluted with buffer containing 300 mM imidazole. The eluted protein was quickly frozen in liquid nitrogen and stored at −80 °C.

**Construction of WT and mutant RyR2 cDNA expression vector and generation of stable HEK293 cell lines.** cDNA for RyR2 was cloned from mouse ventricle and inserted into pcDNA5/FRT/TO vector (ThemoFisher)[35]. Alanine substitution or pathogenic mutations were introduced by inverse PCR using F5

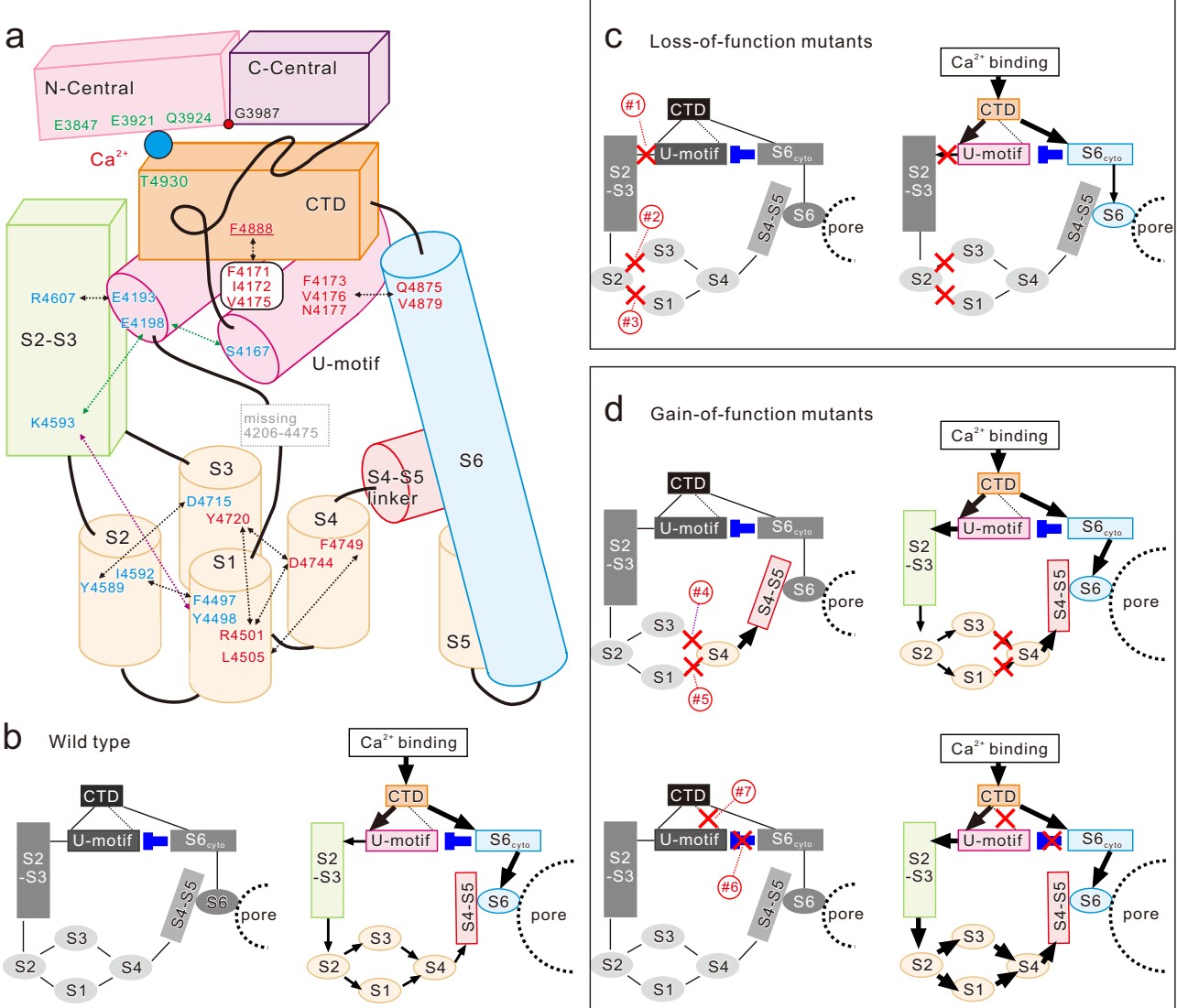

**Fig. 6 Details of interactions and schematic diagram of the channel gating of WT and mutant RyR2 upon Ca²⁺ binding. a** Details of interactions identified in this study. Amino-acid residues shown in red letter and blue letter indicate gain-of-function and loss-of-function by alanine-substituted or pathogenic mutations, respectively. Arrows indicate interactions. Purple and green arrows indicate interactions only found in the closed and open states, respectively. **b–d** Schematic diagram of the channel gating of WT and mutant RyR2 upon Ca²⁺ binding. The left and right diagrams show the states in the absence and presence of Ca²⁺, respectively. The black lines, the domain interactions; The dotted line, CTD/U-motif interaction via F4888; The blue T-shaped lines, U-motif/S6_cyto interaction that acts as a suppressor; The black lines with arrowhead, signals for channel opening. The domains shown in gray-scale and colored indicate the domains in the inactive and active states, respectively. **b** In WT., the channel pore is closed, since the movement of S4-S5 linker is locked. Ca²⁺ binding unlocks the S4-S5 linker and induces the outward leaning of S6, resulting in the pore opening. **c** Loss-of-function (LOF) mutants. Mutations in the U-motif/S2-S3 linker domain (#1), S1/S2 (#2) or S2/S3 (#3) interface cause disconnection of signal transduction. The binding of Ca²⁺ therefore cannot induce the outward movement of the S4-S5 linker and the channel pore is kept closed. **d** Gain-of-function (GOF) mutants. (Upper panels) Mutations in S3/S4 (#4) or S1/S4 (#5) interface unlock the S4-S5 linker to be activated. In the absence of Ca²⁺, the channel pore is kept closed, since the outward leaning of S6 does not occur spontaneously. Binding of Ca²⁺ causes hyperactivity of the channel, since the S4-S5 linker is more active than WT. (Lower panels) Mutations in U-motif/S6_cyto (#6) or CTD/U-motif (#7) reduce or lose U-motif/S6_cyto interaction. The binding of Ca²⁺ causes hyperactivity of the channel, since S6 and the S4–S5 linker are more active than WT.

cassette (nucleotide residue number 10,185–14,901 of mouse RyR2) as a template. The mutations were confirmed by DNA sequencing. HEK293 cells stably expressing WT and mutant RyR2 were generated using the Flp-In T-REx system (ThemoFisher) according to the manufacturer's instructions. Clones with the suitable expression of RyR2 were selected and used for experiments.

**Preparation of microsomes from RyR2-expressing HEK293 cells.** HEK293 cells stably expressing mouse RyR2 (wild type or mutants) were grown in sixty 150-mm cell culture dishes. At 70–80% confluency, protein expression was induced by 2 μg/mL doxycycline (SIGMA) for 48 h. Cells were then harvested, rinsed with cold phosphate-buffered saline (PBS) (GIBCO), and microsomes were prepared as described by Inesi

et al.[39]. Briefly, the cell pellet was resuspended in 60 mL of 10 mM NaHCO₃ in the presence of protease inhibitors and processed for nitrogen cavitation for 30 min at 1000 psi. The suspension was diluted with 60 mL of 0.6 M sucrose, 0.3 M KCl, 40 mM MOPS-Na (pH 7.4), and protease inhibitors and then centrifuged at 1000 × g for 10 min. The supernatant was supplemented with 30 mL of 2.4 M KCl, 0.3 M sucrose, 20 mM MOPS-Na (pH 7.4), and the above protease inhibitor cocktail and centrifuged at 10,000 × g for 20 min. The supernatant was then ultracentrifuged at 100,000 × g for 30 min. The microsomal pellet was resuspended in 60 mL of 0.6 M KCl, 0.3 M sucrose, 20 mM MOPS-Na (pH 7.4), and the protease inhibitor cocktail, and ultra-centrifuged again. Finally, the pellet was resuspended in 12 mL of 0.3 M sucrose, 20 mM MOPS-Na (pH 7.4), and the protease inhibitor cocktail, followed by quick freezing in liquid nitrogen and storage at −80 °C until further use.

**Purification of RyR2**. RyR2 was purified using SBP-FKBP12.6 affinity chromatography[30] with some modifications. The microsomes were solubilized with 2% (w/v) CHAPS (Dojindo) and 1% (w/v) soybean lecithin (Avanti Polar Lipids) in 1 M NaCl, 20 mM MOPS (pH 7.4), 2 mM dithiothreitol, and the protease inhibitor cocktail for 30 min on ice. After centrifugation at $100,000 \times g$ for 30 min at 4 °C, the supernatant was diluted with four volumes of 20 mM MOPS (pH 7.4), 2 mM dithiothreitol, and the protease inhibitor cocktail, after which it was passed through a 0.45-μm filter and loaded onto a pre-equilibrated 5-mL StrepTrap HP column (GE Healthcare, Chicago, IL) with the bound SBP-FKBP12.6 fusion protein. The column was successively washed with 10 column volumes of (1) wash buffer (20 mM MOPS pH7.4, 2 mM dithiothreitol, and 0.3 M sucrose) containing 0.2 M NaCl and 0.25% (w/v) CHAPS and (2) wash buffer containing 0.5 M NaCl and 0.015% (w/v) Tween-20 (Sigma-Aldrich). The SBP-FKBP12.6-RyR2 complex was eluted with the wash buffer supplemented with 2.5 mM D-desthiobiotin (Iba Lifesciences). After checking the purity by SDS-PAGE, the eluate was quickly frozen in liquid nitrogen and stored at −80 °C until further use.

**Negative staining**. The purified RyR2 sample was diluted with a buffer containing 0.2 M NaCl, 20 mM MOPS-Na (pH 7.4), 2 mM dithiothreitol, and 0.015% (w/v) Tween-20, and then applied to pre-hydrophilized carbon-coated EM grids (400 mesh hexagonal copper grids, Stork Veco BV, Netherlands), negatively stained with 1.4% (w/v) uranyl acetate solution, and observed at ×40,000 magnification using a transmission electron microscope (H7500; Hitachi High-Technologies, Tokyo, Japan) operating at 80 kV. Micrographs were taken at ×40,000 using a 1024 × 1024 pixel CCD camera (Fast Scan-F114; TVIPS, Gauting, Germany).

**Cryo-EM sample preparation**. The purified RyR2 sample was buffer-exchanged and concentrated to ~5 mg/mL using Amicon Ultra 100k (Millipore, Burlington, MA) with a buffer containing 0.5 M NaCl, 20 mM MOPS-Na (pH 7.4), 2 mM dithiothreitol, and 0.015% (w/v) Tween-20. EGTA (final concentration of 1 mM) or CaCl₂ (final concentration of 100 μM) was added to the concentrated protein samples to fix the channel to the closed or open state. The concentrated RyR2 was loaded onto a Quantifoil Cu/Rh grid (R1.2/1.3, 300 mesh) (Quantifoil), blotted using Vitrobot Mark III (FEI) with 4 s of blotting time and 100% humidity at 6 °C, and then plunge-frozen in liquid ethane.

**Cryo-EM data collection**. Grids were screened for ice quality, and cryo-EM data were acquired using a Titan Krios G3i cryo-EM (Thermo Fisher Scientific, Waltham, MA) running at 300 kV and equipped with a Gatan Quantum-LS Energy Filter (slit width 25 eV) and a Gatan K3 Summit direct electron detector in the electron counting mode. The electron flux was set to 14 e⁻/pix/s at the detector. For the WT close state with EGTA, imaging was performed at a nominal magnification of ×81,000, corresponding to a calibrated pixel size of 1.07 Å/pix (University of Tokyo, Japan). Electron dose was set to 50 e⁻/Å² for the WT close state with EGTA. For other samples, imaging was performed at a nominal magnification of 105,000×, corresponding to a calibrated pixel size of 0.83 Å/pix. The electron dose was set to 60 e⁻/Å² for the WT open state and 50 e⁻/Å² for K4593A. Each movie was subdivided into 1 e⁻/frame for all datasets. The data were automatically acquired by the image shift method using SerialEM software[40], with a defocus range of −0.8 to −1.6 μm.

**Image processing**. Unless otherwise stated, the same procedure was used to process the data. The movie frames were motion corrected by RELION and CTF parameters and estimated by CTFFIND4[41]. All the following processes were performed using RELION (ver. 3.0 and 3.1)[42]. To generate 2D templates for automatic particle picking, particles were picked from 100 randomly selected micrographs using template-free Laplacian-of-Gaussian picking, then subjected to multiple rounds of reference-free 2D classification. Good 2D classes were selected as templates and 2D template-based particle picking was performed. Picked particles were extracted with 4× down-sampling and subjected to one round of 2D classification. Selected good particles from the 2D classification were submitted to the 3D classification. After one round of 3D classification, good particles were selected, re-extracted with 1.5× down-sampling, and subjected to 3D refinement. The resulting 3D map and particle set were then subjected to beam-tilt and per-particle defocus refinement, Bayesian polishing, and a second round of per-particle defocus refinement followed by 3D refinement. To separate the particles based on open/close states, the no-align 3D classification was performed using a mask covering the TM region. Good classes were subjected to final 3D refinement and postprocessing. The effective resolutions were determined according to the Fourier shell correlations (FSC) = 0.143 criterion. Detailed information is listed in Supplementary Table 1 and described in Figs. S2, S13, and S14.

**Model building, refinement, and analysis**. Model building was performed using COOT[43]. RyR1 in the closed state (PDB accession code, 5TB0 [https://doi.org/10.2210/pdb5TB0/pdb]) was used as the initial reference model. The coordinates were rigid-body fitted in UCSF Chimera[44]. After substitutions in mouse RyR2 sequence and manual building of the model, real space refinement was performed with PHENIX[45,46] with secondary structure and geometry restrained. The residue

sequences (1–10, 85–108, 861–864, 954–969, 1015–1026, 1063–1083, 1275–1283, 1447–1565, 1851–1890, 2010–2055, 2362–2378, 2443–2451, 2659–2711, 2757–2760, 2785–2834, 2906–2915, 2943–2960, 3030–3103, 3130–3135, 3221–3230, 3435–3476, 3580–3610, 3649–3658, 3700–3711, 4206–4272, 4312–4477, 4522–4555, and 4963–4966) were omitted, as the corresponding densities were not visible in all of the maps. All figures were prepared using PyMOL (The PyMOL Molecular Graphics System; http://www.pymol.org). Pore radii along the ion conducting pathway were calculated with HOLE[47]. Buried surface areas were calculated with CNS[37].

**[³H]Ryanodine binding**. [³H]Ryanodine binding assay was carried out as described previously[9,48]. Briefly, microsomes prepared from HEK293 cells expressing RyR2 were incubated for 1 h at 25 °C with 5 nM [³H]ryanodine (PerkinElmer) in reaction media containing 0.17 M NaCl, 20 mM MOPSO-Na (pH 7.0), 2 mM dithiothreitol, 1 mM AMP, and 1 mM MgCl₂. Free Ca²⁺ was adjusted with 10 mM EGTA using WEBMAXC STANDARD, a program on the web (https://somapp.ucdmc.ucdavis.edu/pharmacology/bers/maxchelator/webmaxc/webmaxcS.htm). The [³H]ryanodine binding data ($B$) were normalized to the maximum number of functional channels ($B_{max}$), which was separately determined by Scatchard plot analysis using various concentrations (3–20 nM) of [³H]ryanodine in a high-salt medium. The resultant $B/B_{max}$ represents the averaged activity of each mutant.

**Single-cell Ca²⁺ imaging**. Single-cell Ca²⁺ imaging was performed using HEK293 cells expressing WT or mutant RyR2[9,35]. Ca²⁺ signals from the cytoplasm ([Ca²⁺]_cyt) and ER lumen ([Ca²⁺]_ER) were monitored with G-GECO1.1 (a gift from Robert Campbell from the University of Alberta; Addgene plasmid #32445)[49]. and R-CEPIA1er (a gift from Masamitsu Iino, Nihon University, Tokyo, Japan; Addgene plasmid #58216)[50], respectively. Cells were transfected with cDNAs for these Ca²⁺ indicators 26–28 h before measurement, and at the same time, doxycycline (2 μg/mL) was added to the culture medium to induce RyR2 expression. Ca²⁺ signals were measured in HEPES-buffered Krebs solution (140 mM NaCl, 5 mM KCl, 2 mM CaCl₂, 1 mM MgCl₂, 11 mM glucose, and 5 mM HEPES at pH 7.4) for 5 min and then in 10 mM caffeine-containing Krebs solution for 1.5 min. At the end of each experiment, $F_{min}$ and $F_{max}$ were obtained with 0Ca-Krebs solution containing 20 μM ionomycin, 5 mM BAPTA, and 20 μM cyclopiazonic acid and 20Ca-Krebs solution containing 20 μM ionomycin and 20 mM CaCl₂, respectively[35]. The fluorescence signal ($F − F_{min}$) was normalized to the maximal fluorescence intensity ($F_{max} − F_{min}$). Measurements were carried out at 26 °C.

**Statistics**. Data are presented as means ± SD. One-way analysis of variance (ANOVA), followed by Dunnett's multiple comparisons test, was performed to compare the multiple groups. Statistical analysis was performed using Prism v9 (GraphPad Software, Inc., La Jolla, USA).

**Reporting summary**. Further information on research design is available in the Nature Research Reporting Summary linked to this article.

## Data availability

All data generated or analyzed in this study are available within the article and its Supplementary Information. All raw data supporting the findings of this study are available from the corresponding author upon request. Atomic coordinates and cryo-EM density maps have been deposited in the Protein Data Bank (PDB) and the Electron Microscopy Data Bank (EMDB) under the accession codes 7VML [https://doi.org/10.2210/pdb7VML/pdb] and EMD-30688 (closed state before classification), 7VMM [https://doi.org/10.2210/pdb7VMM/pdb] and EMD-30689 (closed state class1), 7VMN [https://doi.org/10.2210/pdb7VMN/pdb] and EMD-30690 (closed state class2), 7VMO [https://doi.org/10.2210/pdb7VMO/pdb] and EMD-30691 (open state class1), 7VMP [https://doi.org/10.2210/pdb7VMP/pdb] and EMD-30692 (open state class2), 7VMQ [https://doi.org/10.2210/pdb7VMQ/pdb] and EMD-30693 (open state class3), 7VMR [https://doi.org/10.2210/pdb7VMR/pdb] and EMD-32036 (K4593A mutant in the presence of 1 mM EGTA), 7VMS [https://doi.org/10.2210/pdb7VMS/pdb] and EMD-32037 (K4593A mutant in the presence of 100 μM of Ca²⁺). Source data are provided with this paper.

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

## Acknowledgements

We thank Mariko Kurakata for her assistance with cell culture. We thank Ikue Hiraga and the Laboratory of Radioisotope Research, Research Support Center, Juntendo University Graduate School of Medicine, for technical assistance. We also thank the staff scientists at the University of Tokyo's cryo-EM facility. This study was partly supported by the Japan Society for the Promotion of Sciences KAKENHI (grant numbers 19K07105 and 22K06652 to N.K., 19H03404 and 22H02805 to T.M. and JP16H04748 and 21H02411 to H.O.); the Platform Project for Supporting Drug Discovery and Life Science Research (Basis for Supporting Innovative Drug Discovery and Life Science Research [BINDS]; grant number JP20am0101080 to H.O. and T.M. and JP19am0101115 (support number 0064)); the Practical Research Project for Rare/Intractable Diseases from the Japan Agency for Medical Research and Development (AMED; grant number 19ek0109202) to N.K.); an Intramural Research Grant (2–5) for Neurological and Psychiatric Disorders of NCNP (to T.M.); and the Vehicle Racing Commemorative Foundation (6114, 6237, and 6303) to T.M. and H.O.

## Author contributions

T.K., N.K., T.M., and H.O. conceived and designed the project. M. Kodama and H.O. performed cell culture. T.K. performed protein purification. K.S. performed negative-staining EM studies. A.T. and T.K. prepared the grid for cryo-EM. A.T. and M. Kikkawa processed the images. H.O. performed model building and refinement. T.M. and N.K. performed the functional analysis. T.K., A.T., N.K., K.S., M. Kodama, T.S., M. Kikkawa, T.M., and H.O. interpreted the data. T.M. and H.O. supervised the project. H.O. and T.M. wrote the manuscript with input from all authors. All authors reviewed the results and approved the final version of the manuscript.

## Competing interests

The authors declare no competing interests.
