## [Peer Review File · Nature Communications]

Molecular basis for gating of cardiac ryanodine receptor explains the mechanisms for gain- and loss-of function mutationsREVIEWER COMMENTS

Reviewer #1 (Remarks to the Author):

This is an elegant and thorough structure/function study aimed at getting further insight into the molecular mechanism of RyR2 opening. The 3D structures of RyR2 wt K4593A in open and closed conditions enable to resolve inter-domain interactions; which are then tested using Ryanodine binding and Ca²⁺ imaging. The 3D structures of RyR2 K4593A in open and closed conditions were also solved. The main result is the identification and validation of a novel indirect allosteric pathway that reaches the pore of the channel. It involves the U-motif, the S2-S3 linker domain, the voltage-sensor domain, and the S4-S5 linker. The high number of validation mutations, more than 40, including more than a dozen disease mutations, allows comparative functional analysis. The structural and functional study allows to propose a new hypothesis explains both GOF and LOF mutations in the region. The figures are appropriate and illustrate all the main points; and Figure 6 places all the mutations in the structural context. The movies are also very helpful.

Overall, this big effort to analyze the movements of the channel will undoubtedly advance the field of RyR and ion channels in general. In summary, I believe that this work would be an excellent contribution to Nature Communications. My comments to improve the manuscript are below.

MAIN CONCERN

This is a highly specialized subject of a protein that does not benefit from the extensive “common knowledge” that other ion channels have. In addition, it is a huge channel with many domains, and the jargon is unavoidable. Therefore, for each results section I would recommend adding few sentences to offer a parallel, easier lecture to readers less familiar in the field.

SPECIFIC POINTS

The validation files show a high clash score. Although this does not affect the region being examined, with the resolution level it should be possible to achieve better statistics.

The authors indicate that the different 3D reconstructions coming from the same conditions are quite consistent among them. A supplementary figure illustrating reproducibility of their electron densities for the most relevant residues would constitute an additional tool to cross-validate the findings.

The 3D mutant that has been determined is K4593A, however the mutant described in methods is F4888A

The “S2-S3 domain” term is confusing as it includes two TM helices. It would be more accurate to refer to it as S2-S3 linker domain or a similar term.

Line 132: “WT RyR2 exhibited biphasic Ca^{2+} -dependent [3H]ryanodine binding...”

Please explain that this is the normal 3HRy binding profile for the more general audience.

Line 133 “Two pathogenic mutations, K4593Q and K4593R, also led to loss of binding.”

The K to R mutation is very conservative which illustrates the subtlety of RyR’s domain interfaces. It would be helpful that the authors elaborate/discuss this unexpected LOF effect.

Line 134: “Similarly, binding was severely reduced after alanine substitutions (Y4498A, S4167A, and E4198A) and pathogenic mutation (S4167P) in interacting partners (Fig. 2d-e and Extended Data Fig. 3b).”

In this sentence, the term “interacting partners” may evoke interaction with other protein partners; please improve sentence.

Lines 152, section “Movements of the S1-S4 bundle lead to outward movement of the S4-S5 linker”

The finding that mutations affecting the interactions between different helices of the voltage-sensor like domain can either activate or inhibit is unexpected but real. The authors should expand on the differential role of the helical bundle helices; currently it is not very clear.

Line 192 “In the absence of Ca^{2+} , the channel is stabilized in the closed state as the S4-S5 linker is locked by the “stopper,” in form of L4505-F4749 interaction, which prevents α -helix formation of the upper part of S4.”

The concept of “stopper” is very relevant, and should be described in more detail. Here are some ideas: Clarify how the interaction prevents alpha helix formation. In Fig. 3, label the stopper feature. Some interactions cannot release the stopper, resulting in LOF, while others do the opposite, resulting in GOF. The functional results are already relevant per se. The authors seem to envision a mechanism, but this should be made more transparent to the reader.

Line 300: “In addition, S6cyto restricts movement of the U-motif toward the S2-S3 domain.”

It is counterintuitive that S6cyto affects the U-motif, as it appears that S6cyto is both the cause and the consequence of the movement of the U-motif. Please explain better.

Line 304: "However, since S6cyto still restricts the U-motif, the activity may be medium (Fig. 6b, right)."

I assume that "medium" is equivalent to the level of opening of WT? This should be pointed out.

Line 306: "Upon Ca²⁺ binding, conformational changes stop on the way, resulting in loss-of-function (Fig. 6c, right)."

Stopping on the way sounds odd, please improve.

Lines 318-328. In this paragraph of the discussion, the open structure in the presence of only Ca²⁺ is compared to the open structures with Ca²⁺ and either PCB95 or caffeine added. The paragraph is purely descriptive and would be enhanced with discussion on at least what could be the effect of caffeine, whose location is known. The sentence "Ca²⁺ 326 /PCB95 open structure was only determined in the absence of FKBP12.6; addition of FKBP12.6 closes the channel" is confusing; please re-write.

Lines 346-354: The stopper feature should be re-defined here in the discussion.

Figure 6a: yellow arrows are not visible, maybe this refers to the green arrows? What does the green coloring for the residues in the N-central and CTD mean?

Signed by Montserrat Samso

Reviewer #2 (Remarks to the Author):

The manuscript by Kobayashi et al. studied the gating mechanism of RyR2 by Ca²⁺. The authors presented two high-resolution structures of mouse RyR2 in both closed and open states and used many point mutations to test their hypothesis based on whether the mutations caused gain- or loss-of-function. The authors described two independent pathways for RyR2 gating. The first involves interactions close to the channel pore, which are destabilized upon Ca²⁺ binding. The second pathway include a series of conformational transductions from U-motif to S4-S5 linker, which moves out of the way and opens the gate.

This is the first paper to solve the full-length mouse RyR2 structure in both closed and Ca²⁺ bound open states, both WT and K4593A mutant. This is also the first paper trying to explain the gating of RyR2 in detail using structural approach. The study is important and meaningful; the manuscript is well-written, but I feel that some clarification is needed and some small mistakes need to be corrected. Specifically:

1. Could the authors explain in more detail about what is the cause of the U-motif compaction? In my understanding, the direct conformational changes upon Ca²⁺ binding are in the CTD since it contains the Ca²⁺ binding site. The only interaction between CTD and U-motif seems to be the hydrophobic interaction between F4888 in the CTD and the hydrophobic pocket in the U-motif. However, in K4593A mutant, there is no U-motif compaction/rotation. The authors described the CTD/U-motif/S6cyto complex to be “tightly attached to each other” (line 108). If so, why in the K4593A mutant there is no rotation of the complex but rather only the CTD is moved after Ca²⁺ binding? The authors explained that the K4593-S4167 interaction is important for U-motif compaction. It seems to me that the interaction is not the cause of the compaction, instead, it stabilizes the compaction. Could the authors describe how the conformational changes transduce from Ca²⁺ binding domain of the CTD to the U-motif compaction?

2. In data method, the authors described the data collection and image processing of the mutant F4888A. However, the authors showed no structure of F4888A in the paper. And the data collection and image processing details for K4593A is missing in the Methods section. This needs to be corrected.

3. With regarding to F4888A, I think it is actually very interesting to see the structure if the authors indeed collected and processed the data. To my understanding, this mutation will abolish the interaction between CTD and the U-motif, which means it will stop the conformational transduction from CTD downstream. The result, I imagine, should be similar to the structure of K4593A where only the CTD moved upon Ca²⁺ binding. However, functional study proved that it is actually a GOF mutation. Could the authors explain in more detail how the destabilization of CTD-Umotif interaction causes GOF? Are there other interactions between CTD and the U-motif? If the authors can show more interaction between CTD and U-motif that would be really helpful to understand the GOF of F4888A mutant and the alanine-substitutions of the neighboring residues that forms the hydrophobic gate in the U-motif.

4. In Fig6, for panel A legend, I believe the authors made a mistake stating that “Yellow and dark blue arrows indicate interactions only found in the closed and open states, respectively”. Purple arrows seem to indicate closed-state-only interactions while green arrows seem to indicate open-state-only interactions. Also, could the authors show and discuss the case of the alanine-substitution of F4888 and neighboring residues in panel D?

5. In line 234, the figure referred to should be extended fig 5g. Also in line 240, the figure referred to should be extended fig 5g.

6. In movie S4, D4715 is mislabeled as "D4751".

7. Whenever we encounter a LOF mutation, there is always a possibility that the channels are misfolded/did not go to the membrane. Could the authors prove that all the LOF mutants that did not show any signal in the ryanodine binding assays are correctly folded channels? If not, could the authors explain why they think it is not necessary or not possible prove it?

8. Could the authors explain why different 3D classes were found during Cryo-EM data processing if "no major differences were observed among the classes"? (Line91)

9. It would be helpful if the authors could show the distances between atoms when highlighting interactions in atomic models.

10. The authors hypothesized that there are two independent pathways for channel opening. If the authors could make a double GOF mutant from each of the pathways (e.g. D4744A and V4879) and use functional experiments to show that there is an additive effect or even spontaneous channel opening, that would be interesting and would make the argument stronger.

Response to Reviewer's comments:

We really appreciate the positive, very careful reading and constructive comments. We have responded to all the reviewers' comments and incorporated all the suggestions. All changes in the manuscript text file were marked in yellow highlighting.

Major revisions in the revised manuscript are as follows.

1. At the beginning of each results section, some explanations to clarify which part described in the overall movement were added for the understanding to readers less familiar in the field.
2. Density maps for the most relevant residues in different 3D reconstructions coming from the same conditions were attritinary presented in the Supplementary Figures 3b, 4d, h, 6a.
3. Detailed description of biphasic Ca^{2+} -dependent [^3H]ryanodine binding was added.
4. Detailed description about K4593R causing LOF, in spite that K to R is thought to be generally mild mutation, was added.
5. Modification of Figure 3o to explain the role of a "stopper" formed by the interaction of L4505-F4749, and detailed description about the "stopper" formed by the interaction of L4505-F4749 was added.
6. Detail description for an explanation of the mechanism, by which U-motif compaction occurs, was added.
7. Addition of Supplementary Figure 9f and detailed description for the reason why only CTD rotated upon Ca^{2+} binding in K4593A mutant.
8. Modification of Fig. 6b-d, and detailed description for the role of F4888 as a negative suppressor.
9. Addition of a description to confirm that loss-of-function mutants, K4593A and other substitutions, formed functional channels.
10. Addition of all distances of all highlighting interactions in atomic models.

Detailed responses to the reviewers' comments are below.

Reviewer #1 (Remarks to the Author):

This is an elegant and thorough structure/function study aimed at getting further insight into the molecular mechanism of RyR2 opening. The 3D structures of RyR2 wt K4593A in open and

closed conditions enable to resolve inter-domain interactions; which are then tested using Ryanodine binding and Ca²⁺ imaging. The 3D structures of RyR2 K4593A in open and closed conditions were also solved. The main result is the identification and validation of a novel indirect allosteric pathway that reaches the pore of the channel. It involves the U-motif, the S2-S3 linker domain, the voltage-sensor domain, and the S4-S5 linker. The high number of validation mutations, more than 40, including more than a dozen disease mutations, allows comparative functional analysis. The structural and functional study allows to propose a new hypothesis explains both GOF and LOF mutations in the region. The figures are appropriate and illustrate all the main points; and Figure 6 places all the mutations in the structural context. The movies are also very helpful. Overall, this big effort to analyze the movements of the channel will undoubtedly advance the field of RyR and ion channels in general. In summary, I believe that this work would be an excellent contribution to Nature Communications. My comments to improve the manuscript are below.

MAIN CONCERN

This is a highly specialized subject of a protein that does not benefit from the extensive “common knowledge” that other ion channels have. In addition, it is a huge channel with many domains, and the jargon is unavoidable. Therefore, for each results section I would recommend adding few sentences to offer a parallel, easier lecture to readers less familiar in the field.

Thank you for your kind advice. At the beginning of each results section, we added some explanations to clarify which part we describe in the overall movement (lines 158-159, 181-182, 214, 262-263).

SPECIFIC POINTS

The validation files show a high clash score. Although this does not affect the region being examined, with the resolution level it should be possible to achieve better statistics.

Thank you very much for your point out. We deeply understand the importance of the achievement of better statistics. We notice that clashscore of our data is increased by a low resolution of a large N-terminal cytoplasmic region, which is not focused on in this paper (Supplemental Fig. 1f, g, 8a, b). Therefore, we calculated the clashscore for the core domain by excluding the cytoplasmic region and found that they became lower than those for the whole molecule.

We added these values to Supplementary Table 1. In addition, we would like to point out that according to the recent paper published by Dr. Adams's group, the development team of the refinement program PHENIX (*Acta Cryst.* **D77**, 48-61, 2021; <https://pubmed.ncbi.nlm.nih.gov/33404525/>), clashscore around 10 is not so high among 2,750 molecules deposited in the PDB that have map resolutions of better than 5 Å (see Fig. 6a in the paper). We will continue to make efforts to achieve better statistics and will update the PDB as soon as they are corrected.

The authors indicate that the different 3D reconstructions coming from the same conditions are quite consistent among them. A supplementary figure illustrating reproducibility of their electron densities for the most relevant residues would constitute an additional tool to cross-validate the findings.

We appreciate your constructive suggestion. We now added supplementary figures illustrating the reproducibility of their density maps for the most relevant residues in different 3D reconstructions coming from the same conditions (Supplementary Fig. 3b, 4d, 4h, 6a).

The 3D mutant that has been determined is K4593A, however the mutant described in methods is F4888A

We apologize for this mistake. It is now corrected (line 499).

The "S2-S3 domain" term is confusing as it includes two TM helices. It would be more accurate to refer to it as S2-S3 linker domain or a similar term.

We changed "S2-S3 domain" to "S2-S3 linker domain".

Line 132: "WT RyR2 exhibited biphasic Ca^{2+} -dependent [3H]ryanodine binding... "Please explain that this is the normal 3HRy binding profile for the more general audience.

We added a detailed description as follows: "This is explained by two independent Ca^{2+} binding sites; binding of Ca^{2+} to high-affinity site formed by N-Central and CTD opens the channel,

whereas that to undetermined low-affinity site closes the channel (Murayama & Kurebayashi, 2011; Meissner, 2017) (lines 131-133).

Line 133 “Two pathogenic mutations, K4593Q and K4593R, also led to loss of binding.” The K to R mutation is very conservative which illustrates the subtlety of RyR’s domain interfaces. It would be helpful that the authors elaborate/discuss this unexpected LOF effect.

Thank you very much for your indication. As you pointed out, K to R mutation is generally thought to be conservative mutation. However, in K4593R, two nitrogen atoms (N η 1 and N η 2) at the tip of Arg might occupy both oxygen atoms (O ϵ 1 and O ϵ 2) from E4198 in the open state, which prevents S4167-E4198 interaction essential for the U-motif compaction and the channel opening. Therefore, it is reasonable that K4593R exhibited similar behavior as observed in K4593A.

To explain these, we added a sentence “In K4593R, two nitrogen atoms (N η 1 and N η 2) at the tip of Arg might occupy both oxygen atoms (O ϵ 1 and O ϵ 2) from E4198 in the open state, which prevents S4167-E4198 interaction essential for the channel opening.” after the sentence “Two pathogenic mutations, K4593Q and K4593R, also led to loss of binding.” (lines 135-138)

Line 134: “Similarly, binding was severely reduced after alanine substitutions (Y4498A, S4167A, and E4198A) and pathogenic mutation (S4167P) in interacting partners (Fig. 2d-e and Extended Data Fig. 3b).” In this sentence, the term “interacting partners” may evoke interaction with other protein partners; please improve sentence.

We revised the term “interacting partners” to “the interacting pairs” (line 139).

Lines 152, section “Movements of the S1-S4 bundle lead to outward movement of the S4-S5 linker” The finding that mutations affecting the interactions between different helices of the voltage-sensor like domain can either activate or inhibit is unexpected but real. The authors should expand on the differential role of the helical bundle helices; currently it is not very clear.

We changed the description to clearly explain the differential role of the TM helices by adding a new figure (Fig. 3o) (lines 207-211).

Line 192 *“In the absence of Ca²⁺, the channel is stabilized in the closed state as the S4-S5 linker is locked by the “stopper,” in form of L4505-F4749 interaction, which prevents α -helix formation of the upper part of S4.” The concept of “stopper” is very relevant, and should be described in more detail. Here are some ideas: Clarify how the interaction prevents alpha helix formation. In Fig. 3, label the stopper feature. Some interactions cannot release the stopper, resulting in LOF, while others do the opposite, resulting in GOF. The functional results are already relevant per se. The authors seem to envision a mechanism, but this should be made more transparent to the reader.*

Thank you very much for your creative suggestion. Rewinding of the upper part of S4 and S4-S5 linker to an α -helix are linked, and neither form α -helix in the closed state, but both form α -helix in the open state. Since α -helix formation of the S4-S5 linker shortened its length in the open state (Fig. 4o, Supplementary Fig. 4g), we assume that the rewinding to α -helix in the upper part of S4 provides a margin for shortening of the S4-S5 linker. Therefore, it is reasonable that the stopper (L4505-F4749 interaction) is required to stabilize the unwinding form of the upper part of S4 and S4-S5 linker.

To explain these, we replaced Fig. 3o with the new one. In addition, the following sentences are added to the main text. *“Since α -helix formation of the S4-S5 linker shortened its length in the open state (Fig. 3o, Supplementary Fig. 4g), we assume that the rewinding to α -helix in the upper part of S4 provides a margin for shortening of the S4-S5 linker.”* (lines 205-208)

Line 300: *“In addition, S6_{cyto} restricts movement of the U-motif toward the S2-S3 domain. “ It is counterintuitive that S6_{cyto} affects the U-motif, as it appears that S6_{cyto} is both the cause and the consequence of the movement of the U-motif. Please explain better.*

We changed it to *“In addition to direct regulation of the channel gate, S6_{cyto} interacts with U-motif to restrict its movement toward the S2-S3 linker domain (Fig. 6b, left).”* (lines 327-329).

Line 304: *“However, since S6_{cyto} still restricts the U-motif, the activity may be medium (Fig. 6b, right).” I assume that “medium” is equivalent to the level of opening of WT? This should be pointed out.*

We changed it to “Since the interaction still maintains at the open state, the WT channel is not fully activated”. (lines 330-331).

Line 306: "Upon Ca²⁺ binding, conformational changes stop on the way, resulting in loss-of-function (Fig. 6c, right)." Stopping on the way sounds odd, please improve.

We changed it to “This interrupts Ca²⁺-induced conformational changes, resulting in loss-of-function”. (lines 333-334).

Lines 318-328. In this paragraph of the discussion, the open structure in the presence of only Ca²⁺ is compared to the open structures with Ca²⁺ and either PCB95 or caffeine added. The paragraph is purely descriptive and would be enhanced with discussion on at least what could be the effect of caffeine, whose location is known.

We changed it to “In the Ca²⁺/ATP/caffeine structure, in contrast, both CTD, U-motif and S2-S3 domain made a translational outward movement toward caffeine (Supplementary Fig. 10a, middle).” (lines 349-351)

The sentence “Ca²⁺ /PCB95 open structure was only determined in the absence of FKBP12.6; addition of FKBP12.6 closes the channel” is confusing; please re-write.

We changed it to “Ca²⁺/PCB95 open structure was only determined in the absence of FKBP12.6, because the addition of FKBP12.6 closes the channel.” (lines 353-354)

Lines 346-354: The stopper feature should be re-defined here in the discussion.

We described that “We showed that the corresponding salt bridges between R4501, Y4720, and D4744, which are critical to keep S4 in place for appropriate positioning of the stopper (L4505-F4749 interaction), were maintained in both closed and open states (Fig. 3 and Supplementary Fig. 4).”. (lines 375-377)

Figure 6a: yellow arrows are not visible, maybe this refers to the green arrows? What does the green coloring for the residues in the N-central and CTD mean?

We apologize for this mistake. It is now corrected to purple and green.

Signed by Montserrat Samsó

Reviewer #2 (Remarks to the Author):

The manuscript by Kobayashi et al. studied the gating mechanism of RyR2 by Ca²⁺. The authors presented two high-resolution structures of mouse RyR2 in both closed and open states and used many point mutations to test their hypothesis based on whether the mutations caused gain- or loss-of-function. The authors described two independent pathways for RyR2 gating. The first involves interactions close to the channel pore, which are destabilized upon Ca²⁺ binding. The second pathway include a series of conformational transductions from U-motif to S4-S5 linker, which moves out of the way and opens the gate. This is the first paper to solve the full-length mouse RyR2 structure in both closed and Ca²⁺ bound open states, both WT and K4593A mutant. This is also the first paper trying to explain the gating of RyR2 in detail using structural approach. The study is important and meaningful; the manuscript is well-written, but I feel that some clarification is needed and some small mistakes need to be corrected. Specifically:

1. Could the authors explain in more detail about what is the cause of the U-motif compaction? In my understanding, the direct conformational changes upon Ca²⁺ binding are in the CTD since it contains the Ca²⁺ binding site. The only interaction between CTD and U-motif seems to be the hydrophobic interaction between F4888 in the CTD and the hydrophobic pocket in the U-motif. However, in K4593A mutant, there is no U-motif compaction/rotation. The authors described the CTD/U-motif/S6_{cyto} complex to be “tightly attached to each other” (line 108). If so, why in the K4593A mutant there is no rotation of the complex but rather only the CTD is moved after Ca²⁺ binding? The authors explained that the K4593-S4167 interaction is important for U-motif compaction. It seems to me that the interaction is not the cause of the compaction, instead, it stabilizes the compaction. Could the authors describe how the conformational changes transduce from Ca²⁺ binding domain of the CTD to the U-motif compaction?

[Answer to “What is the cause of the U-motif compaction?”]

Thank you very much for your creative suggestion. Upon binding of Ca²⁺ to CTD, CTD and the upper part of S6_{cyto} rotated. Along with rotation, the upper part of S6_{cyto} pushes the N-terminal helix of U-motif, thereby, the compaction of U-motif occurred. This compaction is stably supported because of the key interactions between K4593-E4198-S4167. The compaction of the U-motif creates a space in which the S6_{cyto} can move further, eventually leading to the outward leaning of the S6. That is, the movement of S6_{cyto} and the compaction of U-motif are in a complementary relationship.

To explain the above motions more precisely, the following sentences are added in the section “U-motif plays a key role in stabilizing the channel in the closed state”.

“Upon binding of Ca^{2+} to CTD, the upper part of S6_{cyto} rotated together with CTD (Fig. 1g, 4g). Along with the rotation, S6_{cyto} pushed the N-terminal helix of U-motif, thereby, the parallel shift of the N-terminal helix occurred.” (lines 229-231)

We also found mistakes in the direction of arrows Fig. 4g, 5h, Supporting Fig. 6f (right). In all cases, the direction of S6_{cyto} should be slightly oriented toward the N-terminal helix of U-motif, therefore, we corrected them.

[Answer to “*why in the K4593A mutant there is no rotation of the complex but rather only the CTD is moved after Ca^{2+} binding?*”]

CTD is placed between 1-turn β sheet and U-motif (Supplementary Fig. 9f). Since U-motif, CTD, and S6_{cyto} rotated together upon Ca^{2+} binding, it is reasonable that structures of these regions in closed and open states matched well (Supplementary Fig. 9f left).

The independent rotation of CTD in K4593A indicates that CTD placed between 1-turn β sheet and U-motif has some degree of freedom in the rotation in the K4593A mutant, so that S6_{cyto} , connected just before CTD, the upper part of S6_{cyto} , rotated together with CTD. However, since the compaction of U-motif cannot stably form in K4593A, the rotation cannot proceed further due to collision with the N-terminal helix of U-motif.

To explain more detail about this issue, we added new figures (Supplementary Fig. 9f) and the following sentences at the end of section “Structural basis of the loss-of-function mutation”.

“It is interesting that the rotation due to Ca^{2+} binding occurred only in CTD and the upper part of D6_{cyto} (Fig. 5c, g, Supplementary Fig. 9f). Since CTD is placed between 1-turn β sheet connected just before U-motif, (Supplementary Fig. 9f), U-motif, CTD, and S6_{cyto} are supposed to rotate together upon Ca^{2+} binding as observed in the open state (Fig. 1f, Fig. 5c, g, Supplementary Fig. 9f). The independent movement of CTD in K4593A indicates that CTD placed between 1-turn β

sheet and U-motif have some degree of freedom in the rotation. The upper part of S6_{cyto} rotated together with CTD because of a direct connection to CTD (Supplementary Fig. 9f). However, the rotation throughout S6 was blocked by the collision with the uncompact U-motif in K4593A (Supplementary Fig. 9f).” (lines 290-297)

[For your statement “*The only interaction between CTD and U-motif seems to be the hydrophobic interaction between F4888 in the CTD and the hydrophobic pocket in the U-motif.*”]

We are afraid that you may misunderstand the interaction between CTD and U-motif. We do not state that the hydrophobic interaction between F4888 in the CTD and the hydrophobic pocket in the U-motif is the only interaction between CTF and U-motif. We showed that F4888A lacking this interaction still had activity, and, moreover, no additive effects were observed with N4177A and F4888A (Supplementary Fig. 7b). The result from the double mutant indicates that U-motif/S6_{cyto} and U-motif/CTD interactions are involved in the common pathways, and the interaction between CTD and U-motif via F4888 may work as a negative regulator. To avoid further confusion and to clearly show that the roles of the two interactions are different, we modified the number of the line between CTD and U-motif from one to two in Fig. 6b-d.

We also added the following sentences in the Discussion section.

“In addition to direct regulation of the channel gate, S6_{cyto} interacts with U-motif to restrict its movement toward the S2-S3 linker domain (blue T-shaped line, Fig. 6b, left). This negative regulation by S6_{cyto} is supported by hydrophobic interaction between CTD (F4888) and U-motif (dotted line, Fig. 6b, left). Since the interaction is still maintained at the open state, the WT channel is not fully activated (Fig. 6b, right).” (lines 327-331)

2. In data method, the authors described the data collection and image processing of the mutant F4888A. However, the authors showed no structure of F4888A in the paper. And the data collection and image processing details for K4593A is missing in the Methods section. This needs to be corrected.

We apologize for this mistake. We deleted the description about F4888A and added the description about K4593A in data method section. (line 499)

3. With regarding to F4888A, I think it is actually very interesting to see the structure if the authors indeed collected and processed the data. To my understanding, this mutation will abolish the interaction between CTD and the U-motif, which means it will stop the conformational transduction from CTD downstream. The result, I imagine, should be similar to the structure of K4593A where only the CTD moved upon Ca²⁺ binding. However, functional study proved that it is actually a GOF mutation. Could the authors explain in more detail how the destabilization of CTD-Umotif interaction causes GOF? Are there other interactions between CTD and the U-motif? If the authors can show more interaction between CTD and U-motif that would be really helpful to understand the GOF of F4888A mutant and the alanine-substitutions of the neighboring residues that forms the hydrophobic gate in the U-motif.

Thank you very much for your deep understanding of our structures. We think that your point is correct and important. There is no doubt that the structure of GOF mutant K4888A will contribute to a deeper understanding of the opening mechanism by Ca²⁺ binding. However, we are sincerely afraid that the incorporation of the structure of F4888A into this manuscript made confusion for readers in terms of the volume of text and number of figures. Therefore, we are planning to publish the structure of F4888A in another paper as soon as possible.

4. In Fig6, for panel A legend, I believe the authors made a mistake stating that “Yellow and dark blue arrows indicate interactions only found in the closed and open states, respectively”. Purple arrows seem to indicate closed-state-only interactions while green arrows seem to indicate open-state-only interactions. Also, could the authors show and discuss the case of the alanine-substitution of F4888 and neighboring residues in panel D?

We apologize for this mistake in the legend for Fig. 6a. It is now corrected to purple and green.

We revised Fig. 6b, c, d, and F4888A is now explained as mutation #7 in Fig. 6d.

5. In line 234, the figure referred to should be extended fig 5g. Also in line 240, the figure referred to should be extended fig 5g.

We apologize for this mistake. These are now corrected.

6. *In movie S4, D4715 is mislabeled as “D4751”.*

We apologize for this mistake. It is now corrected.

7. *Whenever we encounter a LOF mutation, there is always a possibility that the channels are misfolded/did not go to the membrane. Could the authors prove that all the LOF mutants that did not show any signal in the ryanodine binding assays are correctly folded channels? If not, could the authors explain why they think it is not necessary or not possible prove it?*

We added a description “10 mM caffeine, a potent RyR activator, released Ca²⁺ from ER in cells expressing K4593A, indicating that it forms a functional channel. These findings confirm a loss-of-function of the K4593A channel. Similar results were obtained with other substitutions in S4167, E4198, and K4593 (Fig. 2g, Supplementary Table 2).” (lines 145-146)

8. *Could the authors explain why different 3D classes were found during Cryo-EM data processing if “no major differences were observed among the classes”? (Line91)*

Different classes during Cryo-EM data processing might correspond to the thermal fluctuations of the molecule. Therefore, it is quite reasonable that there are slight differences among them. Although the structure in the closed state is classified into 2 classes (Supplementary Fig. 1f), all the essential interactions discussed in this manuscript are maintained among them. The above situation was essentially the same in the structure in the open state (Supplementary Fig. 1f). To show that the different 3D reconstructions coming from the same conditions are consistent among them, we now added supplementary figures illustrating reproducibility of their density maps for the most relevant residues (Supplementary Fig. 3b, 4d, h, 6a).

9. *It would be helpful if the authors could show the distances between atoms when highlighting interactions in atomic models.*

In the figures, we added the distances of all highlighting interactions in atomic models.

10. The authors hypothesized that there are two independent pathways for channel opening. If the authors could make a double GOF mutant from each of the pathways (e.g. D4744A and V4879) and use functional experiments to show that there is an additive effect or even spontaneous channel opening, that would be interesting and would make the argument stronger.

Thank you very much for your nice proposal. We are also interested in the two pathways. We are currently testing it by several double mutant channels and planning to present the results with the structures of GOF mutants in the next paper.

REVIEWERS' COMMENTS

Reviewer #1 (Remarks to the Author):

The authors have addressed all my concerns in the new version. Overall, the work constitutes a valuable scientific contribution. I only have a new suggestion and found a minor mistake that was missed in the previous revision.

Line 353: "Ca²⁺/PCB95 open structure was only determined in the absence of FKBP12.6 because the addition of FKBP12.6 closes the channel [26]. This is in contrast to our open structure which binds FKBP12.6".

In support of Kobayashi et al.'s results, a 3D structure of RyR1 open in the presence of Ca²⁺/PCB95/FKBP12 (Samsó et al., 2009) could be mentioned.

Samsó, M., W. Feng, I.N. Pessah, and P.D. Allen. 2009. Coordinated Movement of Cytoplasmic and Transmembrane Domains of RyR1 upon Gating. *PLoS Biology*. 7:980-995.

Legend to Supplementary Figure 1d and 1e:

1d: "Ca²⁺ dependence on [3H]ryanodine binding of the purified RyR2 in the presence of 100 μ M Ca²⁺."

However, there is a range of Ca²⁺ concentrations.

1e: "Effect of ATP and caffeine on [3H]ryanodine binding of the purified RyR2."

Concentration of Ca²⁺ is missing, and probably the 100 μ M Ca²⁺ indicated in 1d belongs to 1e.

Reviewer #2 (Remarks to the Author):

The authors corrected their mistakes and answered most of my concerns by adding figure panels and more detailed descriptions in the manuscript. However, I still have some minor concerns, specifically:

1. The sentence in 216-217, “We found a close contact between U-motif and S6cyto as observed in the closed state structure of RyR1 (Fig. 4a, b Supplementary Fig. 6a and Movie S5)” is very confusing. From the wording of this sentence, Fig. 4a seems to be describing the structure of RyR1. Please reword this sentence to make it more clear and easier for the readers to understand. Also, please make sure all the figures and figure panels are cited correctly in the manuscript.

2. In Fig 4b and c, it is hard to look at the interaction between F4173 and V4176, or maybe the authors can make the mesh thinner and with lighter color. Also, please delete the “merged” label in Fig 4d. It is confusing since Fig 4b and c have “open” and “closed” labels and the reader will naturally assume that Fig 4d is the merge between Fig 4b and c.

3. The authors should include the S4-S5 linker density maps as a supplementary figure to show that it is indeed a transition from loop to alpha helix because it is such an important part of the mechanism. The density map shown in Supplementary Figure 1 only showed a few residues.

4. It is shown in Supplementary Figure 1C that using 500 mM NaCl greatly increased the channel open probability in the Ryanodine binding assay. And because of this, the authors went on with this condition for protein purification. However, 500 mM NaCl is very far from physiological condition. And since salt concentration does play a role in channel function, do the authors believe the structures obtained in high salt reflect the real situation? Please explain why.

5. In Supplementary Figure 1e, the “no ligand” label is confusing and seems wrong. Please change the label to make it clear.

6. For this manuscript, the functional study is very important, it is a strong evidence to prove and verify the structure results. In Figure 2d, 3d, 4h and 4l, some data points are without error bars. However, in the corresponding figure legends, the authors wrote n=4. Does it mean that some points were only done once? The authors should provide source data containing all the data points and replicas of the functional experiments and upload as a spreadsheet.

7. In movie1, label “S3-S3” should be corrected.

Response to Reviewer's comments:

We appreciate the positive, very careful reading and constructive comments. We have responded to all the reviewers' comments and incorporated all the suggestions. All changes in the manuscript text file were marked in yellow highlighting.

Major revisions in the revised manuscript are as follows.

1. We revised the title as the editorial office suggested.
2. We revised the abstract as the editorial office suggested.
3. We showed density maps overlaid around S4-S5 linker in Supplementary Fig. 4e, f.

Reviewer #1 (Remarks to the Author)

The authors have addressed all my concerns in the new version. Overall, the work constitutes a valuable scientific contribution. I only have a new suggestion and found a minor mistake that was missed in the previous revision.

Line 353: "Ca²⁺/PCB95 open structure was only determined in the absence of FKBP12.6 because the addition of FKBP12.6 closes the channel [26]. This is in contrast to our open structure which binds FKBP12.6".

In support of Kobayashi et al.'s results, a 3D structure of RyR1 open in the presence of Ca²⁺/PCB95/FKBP12 (Samso et al., 2009) could be mentioned.

Samso, M., W. Feng, I.N. Pessah, and P.D. Allen. 2009. Coordinated Movement of Cytoplasmic and Transmembrane Domains of RyR1 upon Gating. PLoS Biology. 7:980-995.

Thank you very much for your suggestion. However, our manuscript mainly deals with cardiac ryanodine receptor (RyR2). We are afraid that, if we discuss skeletal ryanodine receptor (RyR1), the focus will be defused. So, we do not cite the suggested paper here.

Legend to Supplementary Figure 1d and 1e:

1d: "Ca²⁺ dependence on [3H]ryanodine binding of the purified RyR2 in the presence of 100 uM Ca²⁺."

However, there is a range of Ca²⁺ concentrations.

Thank you for pointing it out. We deleted 'in the presence of 100 $\mu\text{M Ca}^{2+}$ '

1e: "Effect of ATP and caffeine on [3H]ryanodine binding of the purified RyR2."

The concentration of Ca^{2+} is missing, and probably the 100 $\mu\text{M Ca}^{2+}$ indicated in 1d belongs to 1e.

Thank you for pointing it out. We added 'in the presence of 100 $\mu\text{M Ca}^{2+}$ '. In addition, we added 'in the presence of 100 $\mu\text{M Ca}^{2+}$ ' in Supplementary Figure 1c.

Reviewer #2 (Remarks to the Author)

The authors corrected their mistakes and answered most of my concerns by adding figure panels and more detailed descriptions in the manuscript. However, I still have some minor concerns, specifically:

1. *The sentence in 216-217, “We found a close contact between U-motif and S6_{cyto} as observed in the closed state structure of RyR1 (Fig. 4a, b Supplementary Fig. 6a and Movie S5)” is very confusing. From the wording of this sentence, Fig. 4a seems to be describing the structure of RyR1. Please reword this sentence to make it more clear and easier for the readers to understand. Also, please make sure all the figures and figure panels are cited correctly in the manuscript.*

Thank you for pointing it out. We changed it to “In the closed state, we found a close contact between U-motif and S6_{cyto} (Fig. 4a, b Supplementary Fig. 6a and Movie S5), which was similar as observed in the closed state of RyR1.”. (lines 224-226)

2. *In Fig 4b and c, it is hard to look at the interaction between F4173 and V4176, or maybe the authors can make the mesh thinner and with lighter color. Also, please delete the “merged” label in Fig 4d. It is confusing since Fig 4b and c have “open” and “closed” labels and the reader will naturally assume that Fig 4d is the merge between Fig 4b and c.*

Thank you for pointing it out. We used a thinner mesh than before. We did not change the color, but we think that the interaction can be clearly seen now. Also. We removed the “merged” label in Fig. 4d.

3. *The authors should include the S4-S5 linker density maps as a supplementary figure to show that it is an indeed a transition from loop to alpha helix because it is such an important part of the mechanism. The density map shown in Supplementary Figure 1 only showed a few residues.*

Thank you for your suggestion. According to your suggestion, we showed density maps overlaid around S4-S5 linker in Supplementary Fig. 4e, f.

4. *It is shown in Supplementary Figure 1C that using 500 mM NaCl greatly increased the channel open probability in the Ryanodine binding assay. And because of this, the authors*

went on with this condition for protein purification. However, 500 mM NaCl is very far from physiological condition. And since salt concentration does play a role in channel function, do the authors believe the structures obtained in high salt reflect the real situation? Please explain why.

In high salt solution, the channel activity is greatly enhanced but Ca^{2+} -dependent activation is preserved (Supplementary Fig. 1c-e). We therefore believe that the fundamental opening mechanism we obtained reflects the real situation. We added descriptions in Results (lines 86-88) and Discussion (lines 360-362).

5. In Supplementary Figure 1e, the “no ligand” label is confusing and seems wrong. Please change the label to make it clear.

Thank you for pointing it out. We changed the label 'no ligand' to ' Ca^{2+} ' and 'ATP/Caf' to ' Ca^{2+} /ATP/Caf'.

6. For this manuscript, the functional study is very important, it is a strong evidence to prove and verify the structure results. In Figure 2d, 3d, 4h and 4l, some data points are without error bars. However, in the corresponding figure legends, the authors wrote n=4. Does it mean that some points were only done once? The authors should provide source data containing all the data points and replicas of the functional experiments and upload as a spreadsheet

In the data points without error bars, the error bars are so small that they are within the symbols. We provided source data containing all the data points.

7. In movie1, label "S3-S3" should be corrected.

We apologize for this mistake. It is now corrected.